# Inhibition of Indoleamine 2,3-Dioxygenase Exerts Antidepressant-like Effects through Distinct Pathways in Prelimbic and Infralimbic Cortices in Rats under Intracerebroventricular Injection with Streptozotocin

**DOI:** 10.3390/ijms25137496

**Published:** 2024-07-08

**Authors:** Yu Qin, Xiao Hu, Hui-Ling Zhao, Nurhumar Kurban, Xi Chen, Jing-Kun Yi, Yuan Zhang, Su-Ying Cui, Yong-He Zhang

**Affiliations:** Department of Pharmacology, School of Basic Medical Sciences, Peking University, Beijing 100191, China; qinyu0309@pku.edu.cn (Y.Q.); huxiao@bjmu.edu.cn (X.H.); zhaohuiling@pku.edu.cn (H.-L.Z.); nunux53@pku.edu.cn (N.K.); 1610305127@pku.edu.cn (X.C.); 1810305301@pku.edu.cn (J.-K.Y.); 1810301314@pku.edu.cn (Y.Z.)

**Keywords:** depression, IDO, prelimbic cortex, infralimbic cortex, astrocyte, microglia

## Abstract

The application of intracerebroventricular injection of streptozotocin (ICV-STZ) is considered a useful animal model to mimic the onset and progression of sporadic Alzheimer’s disease (sAD). In rodents, on day 7 of the experiment, the animals exhibit depression-like behaviors. Indoleamine 2,3-dioxygenase (IDO), a rate-limiting enzyme catalyzing the conversion of tryptophan (Trp) to kynurenine (Kyn), is closely related to depression and AD. The present study aimed to investigate the pathophysiological mechanisms of preliminary depression-like behaviors in ICV-STZ rats in two distinct cerebral regions of the medial prefrontal cortex, the prelimbic cortex (PrL) and infralimbic cortex (IL), both presumably involved in AD progression in this model, with a focus on IDO-related Kyn pathways. The results showed an increased Kyn/Trp ratio in both the PrL and IL of ICV-STZ rats, but, intriguingly, abnormalities in downstream metabolic pathways were different, being associated with distinct biological effects. In the PrL, the neuroprotective branch of the Kyn pathway was attenuated, as evidenced by a decrease in the kynurenic acid (KA) level and Kyn aminotransferase II (KAT II) expression, accompanied by astrocyte alterations, such as the decrease in glial fibrillary acidic protein (GFAP)-positive cells and increase in morphological damage. In the IL, the neurotoxicogenic branch of the Kyn pathway was enhanced, as evidenced by an increase in the 3-hydroxy-kynurenine (3-HK) level and kynurenine 3-monooxygenase (KMO) expression paralleled by the overactivation of microglia, reflected by an increase in ionized calcium-binding adaptor molecule 1 (Iba1)-positive cells and cytokines with morphological alterations. Synaptic plasticity was attenuated in both subregions. Additionally, microinjection of the selective IDO inhibitor 1-Methyl-DL-tryptophan (1-MT) in the PrL or IL alleviated depression-like behaviors by reversing these different abnormalities in the PrL and IL. These results suggest that the antidepressant-like effects linked to Trp metabolism changes induced by 1-MT in the PrL and IL occur through different pathways, specifically by enhancing the neuroprotective branch in the PrL and attenuating the neurotoxicogenic branch in the IL, involving distinct glial cells.

## 1. Introduction

Alzheimer’s disease (AD), a common neurodegenerative disorder, is the most prevalent form of senile dementia, causing progressive deterioration of cognition, behavior, and rational skills. Neuropsychiatric symptoms like depression often accompany and/or precede AD onset [1,2]. Depression is regarded as an early symptom of AD or a risk factor for AD with a negative impact on the quality of life of patients and caregivers [3,4]. However, the pathophysiological mechanisms underlying depression in AD patients remain less well-defined.

Over 95% of AD cases manifest as sporadic AD (sAD), influenced by the intricate interplay of genetic and environmental factors [5]. Streptozotocin (STZ), a glucosamine compound with β-cytotoxic action, is commonly used to induce diabetes in laboratory animals [6]. Some investigators exploited the intracerebroventricular microinjection of STZ (ICV-STZ) to simulate sporadic AD in rodents as a non-transgenic model of this disease and used it for preclinical testing of pharmacological therapies for AD [7]. Single or double ICV-STZ injection(s) chronically produce progressive pathological changes that resemble the molecular, pathological, and behavioral features of AD [8,9,10]. Specifically, ICV-STZ induces amyloid-β (Aβ) accumulation and Tau hyperphosphorylation as a result of oxidative stress and immuno-inflammatory responses in the cortex and hippocampus, which finally cause learning and cognitive impairments [10,11]. Furthermore, ICV-STZ also reproduces AD progression, promoting first non-cognitive features such as depression-like behaviors in rodents [12,13]. Although the pathophysiological mechanism of depression-like behaviors in the STZ-induced AD model is unclear, some pilot research indicated that the activation of indoleamine-2,3-dioxygenase (IDO) might be a key factor [14].

IDO is a rate-limiting enzyme in the kynurenine pathway that catalyzes the conversion of tryptophan (Trp) to kynurenine (Kyn) [15]. Subsequently, Kyn is metabolized by kynurenine aminotransferase (KAT) II in astrocytes, and converted to kynurenic acid (KA), which exerts neuroprotective effects. In addition, Kyn is metabolized by kynurenine 3-monooxygenase (KMO) in microglia, and its metabolites, 3-hydroxykynurenine (3-HK) and quinolinic acid, have the potential to induce excitotoxic effects [15]. The abnormalities in IDO-related Kyn metabolism occurring in glial cells have been reported to play a prominent role in the processes of neuroinflammation [16,17]. Additionally, these abnormalities were also found in ICV-STZ animal models. Souza et al. [18] reported that ICV-STZ-associated depression-like behaviors in mice may be attributable to hippocampal IDO activation in response to the upregulation of innate immune and proinflammatory cytokines. In addition, ICV-STZ caused an acute and persistent neuroinflammatory response, reflected by reactive microgliosis and astrogliosis in the dorsal hippocampus [19] and by the hyperactivation of astrocytes in the prefrontal cortex (PFC) [20]. These results indicated that IDO-related glial alterations in certain brain regions might be involved in the regulation of AD-associated depression.

The prelimbic cortex (PrL) and infralimbic cortex (IL) are two major medial PFC (mPFC) areas in rodents that are thought to mediate the control of depression-like behaviors [21]. Although the PrL and IL have similar projection patterns in certain aspects, a growing body of research has revealed that they can be functionally distinguished with regard to mediating various physiological and behavioral processes in rodents, including fear expression and extinction [22], cocaine-seeking [23], anxiety [24], and depression [25]. These findings bolster the fact that exploring the subregional mechanisms of mPFC is important for understanding the physiological and pathological processes of neurologic disorders.

Therefore, this study investigated depression-like behavior and the neurophysiological mechanism of action of IDO on Kyn pathways in the PrL and IL in the rat model of ICV-STZ. We evaluated the effects of ICV-STZ on the Kyn pathway, astrocytes, microglia, and synaptic plasticity in the PrL and IL. Additionally, the IDO inhibitor 1-MT was directly injected into the PrL and IL to study its antidepressant-like effect and influence on the above pathological mechanisms.

## 2. Results

### 2.1. ICV-STZ Induced Depression- and Anxiety-like Behaviors before Cognitive Impairment

ICV-STZ rats exhibited a significant increase in immobility time in the FST on d 7 and d 14 compared with vehicle rats, indicating depression-like behavior in these rats (Figure 1A). Compared with vehicle rats, immobility time in ICV-STZ rats was not significantly different on d 21 (Figure 1A). In the SPT, ICV-STZ rats exhibited a significant decrease in sucrose preference on d 7 and d 14, indicating anhedonia-like behavior (Figure 1B). In contrast, sucrose preference on d 21 was not significantly different from vehicle treatment (Figure 1B). Furthermore, ICV-STZ rats exhibited a significant decrease in the time spent in the center of the OFT on d 7, d 14, and d 21, indicating anxiety-like behavior (Figure 1C,E). The total distance traveled in the OFT, representing a measure of locomotor activity, was not significantly different among the groups (Figure 1D,E). In the NORT, the exploration time for novel objects significantly increased in relation to control, vehicle, and ICV-STZ 7 d rats, but only a modest, non-significant increase was noted in ICV-STZ 14 d and ICV-STZ 21 d rats (Figure 1F). Additionally, the discrimination ratio was not significantly different between ICV-STZ 7 d rats and vehicle rats, whereas ICV-STZ 14 d and 21 d rats exhibited a significant decrease in the discrimination ratio compared with vehicle rats (Figure 1F). The result from the NORT indicated that impairments in recognition memory and discrimination ability were only observed in ICV-STZ 14 d and 21 d rats.

Next, we examined levels of AD-related proteins in the mPFC in ICV-STZ 7 d rats. No significant differences in the expression of Aβ, Tau, or pTauser199 or the pTauser199/Tau ratio were found in the mPFC in ICV-STZ rats, suggesting that AD-related pathological changes were not predominant in the mPFC on d 7 after ICV-STZ treatment (Figure 1G–K). These results indicated that ICV-STZ induced depression- and anxiety-like behaviors before the expression of cognitive impairment, and as cognition progressively deteriorated, depression-like behavior in ICV-STZ rats gradually disappeared.

### 2.2. ICV-STZ Induced IDO Activation in the PrL and IL

To investigate whether ICV-STZ-induced depression-like behaviors are related to the IDO-mediated kynurenine pathway, we examined the levels of kynurenine metabolites in several brain regions associated with mood disorders, including the anterior cingulate cortex (ACC), PrL, IL, lateral habenula (LHb), dorsal raphe nucleus (DRN), hippocampus (Hip), and locus coeruleus (LC). ICV-STZ significantly increased the Kyn/Trp ratio in the PrL and IL (Figure 2A), which indicated that IDO activity was elevated in these two brain regions when these rats exhibited depression-like behaviors.

Next, to evaluate directly whether the activation of IDO in the PrL and IL is responsible for depression-like behavior in ICV-STZ rats, we subsequently inhibited IDO by microinjecting 1-MT in the PrL or IL and observed its effects on the behaviors of ICV-STZ rats. Cannula placements in the PrL and IL are shown in Figure 2B. There were no significant differences in body weight, food consumption, or water intake for all groups until the end of the experiment, suggesting that ICV-STZ and 1-MT treatment had no effect on energy intake and metabolism (Appendix A). The intra-PrL or intra-IL injection of 1-MT significantly decreased the up-regulation of the Kyn/Trp ratio induced by ICV-STZ (Figure 2C,E). Although ICV-STZ did not cause significant changes in the expression of IDO in the PrL or IL (Appendix A), IDO activity was significantly increased and reversed by the intra-PrL or intra-IL injection of 1-MT (Figure 2D,F). Furthermore, the levels of 5-HT in both the PrL and IL were not significantly altered in response to ICV-STZ or 1-MT (Appendix A).

### 2.3. Intra-PrL or -IL Injection of 1-MT Blocks ICV-STZ-Induced Depression-like Behaviors

Depression-like behaviors that were induced by ICV-STZ, including an increase in immobility time in the FST, a decrease in sucrose preference in the SPT, and an increase in the latency to feed in the NSFT, were reversed by the intra-PrL injections of 1-MT (Figure 3A–C). Food consumption in the NSFT was not significantly different among the groups, excluding ingestion bias (Figure 3D). The reduced time in the center of the OFT was not reversed by 1-MT, and the total distance traveled in the open field was not significantly different among groups (Figure 3E,F).

Similarly, the intra-IL injections of 1-MT reversed ICV-STZ-induced depression-like behaviors such as prolonged immobility time, reduced sucrose preference, and increased latency to feed (Figure 3G–I). In addition, the time in the center of the OFT showed that the intra-IL injections of 1-MT did not reverse anxiety-like behavior in ICV-STZ rats and had no effect on locomotor activity (Figure 3J,K).

These results indicated that ICV-STZ induced depression-like behaviors that were likely related to the activation of IDO in the PrL and IL, and both the intra-PrL and intra-IL injections of 1-MT prevented depression-like behavior in ICV-STZ rats. Additionally, the regionally selective blockade of IDO by 1-MT either in the PrL or IL failed to produce an anxiolytic effect, suggesting that anxiety-like behaviors that are induced by ICV-STZ might not be attributable to IDO activation in the PrL and IL, which needs further confirmation.

### 2.4. ICV-STZ Attenuates Astrocyte Number and Function in the PrL

The kynurenine pathway is regulated by glial cells in the brain. We detected the morphological and functional features of astrocytes and microglia in the PrL and IL to examine the potential antidepressant mechanism of 1-MT. GFAP was used as an astrocyte marker. Iba1 was used as a microglia marker. The results from the PrL and IL are shown in Figure 4 and Figure 5, respectively.

ICV-STZ significantly reduced the number of GFAP+ cells in the PrL, which was reversed by the intra-PrL injection of 1-MT (Figure 4A,B). However, no significant difference was found in the number of Iba1+ cells in the PrL in ICV-STZ rats that were treated with saline or 1-MT (Figure 4A,C). Then, we further analyzed the morphological features of the glia by performing a three-dimensional reconstruction (Figure 4D). Consistent with the above results, the morphological analysis showed a decrease in the surface area of GFAP+ cells, and Sholl analysis showed a lower number of intersections among 5 μm to 40 μm, suggesting the atrophy of astrocytes in the PrL in ICV-STZ rats (Figure 4E,F). The intra-PrL injection of 1-MT reversed the morphological abnormalities induced by ICV-STZ. Additionally, microglial morphology, including terminal points of Iba1+ cells and the soma volume of Iba1+ cells in the PrL, was not affected by either ICV-STZ or 1-MT (Figure 4G,H).

Next, we examined the expression of glia-related proteins and inflammatory cytokines. The Western blot analysis showed that ICV-STZ markedly decreased the expression of GFAP in the PrL, accompanied by reductions in glutamate–aspartate transporter (GLAST) and glutamate transporter-1 (GLT-1), which are important components of astrocyte function in glutamate transport (Figure 4I–L). These defects were restored by the intra-PrL injection of 1-MT, suggesting that the antidepressant effects of 1-MT in ICV-STZ rats might be attributable to protective effects on astrocytes (Figure 4I–L). On the other hand, the expression of Iba1 in the PrL was not influenced by either ICV-STZ or 1-MT treatment (Figure 4M,N). Inflammatory cytokines, including interleukin 6 (IL-6), interleukin 1β (IL-1β), and tumor necrosis factor-alpha (TNF-α) are secreted by microglia within the brain in response to injury and infection. In the PrL, the levels of IL-1β, IL-6, and TNF-α were not influenced by either ICV-STZ or 1-MT treatment (Figure 4O,P; Appendix A).

### 2.5. ICV-STZ Induces Microglia Activation and Inflammatory Response in the IL

In the IL, ICV-STZ and the intra-IL injection of 1-MT had no effect on the number of GFAP^+^ cells per unit area (Figure 5A,B). Conversely, the number of Iba1^+^ cells in the IL was significantly increased by ICV-STZ, whereas the intra-IL injection of 1-MT recovered the density of Iba1^+^ cells to normal levels (Figure 5A,C).

The morphological characterization of the astrocytes in the IL showed that the area of GFAP^+^ cells did not differ significantly among groups, and Sholl analysis also showed that the number of intersections was not significantly different among groups (Figure 5D–F). In contrast, the analysis of morphological features in microglia showed that ICV-STZ significantly decreased terminal points and increased the soma volume of Iba1^+^ cells in the IL, which are typical morphological features of microglia activation [26,27] (Figure 5G,H). The intra-IL injection of 1-MT prevented these morphological abnormalities of microglia induced by ICV-STZ (Figure 5G,H).

Western blot analysis showed that ICV-STZ had no effects on the expression of GFAP, whereas 1-MT significantly increased GFAP expression in the IL (Figure 5I,J). Additionally, there were no significant differences in the expression of GLAST or GLT-1 among groups (Figure 5K,L). The expression of Iba1 in the IL was significantly increased by ICV-STZ, and this increase was reversed by the intra-IL injection of 1-MT (Figure 5M,N). Correspondingly, the levels of IL-1β and IL-6 in the IL significantly increased in ICV-STZ rats, and the intra-IL injection of 1-MT prevented this inflammatory response, which might be associated with its antidepressant effect (Figure 5O,P). However, the levels of TNF-α in the IL were not influenced by either ICV-STZ or 1-MT treatment (Appendix A).

### 2.6. ICV-STZ Has Different Effects on the Kyn Pathway Branches in the PrL and IL

Kyn and several downstream metabolites have been proposed to be highly correlated with depression [28]. The Kyn pathway branches have been suggested to be separately regulated by microglia and astrocytes, so we next examined Kyn downstream metabolites in both the PrL and IL. 

In the PrL, KA significantly decreased in ICV-STZ rats, and this decrease was reversed by the intra-PrL injection of 1-MT (Figure 6A). However, 3-HK in the PrL was not significantly different among groups (Figure 6B). Furthermore, KAT II expression in the PrL in ICV-STZ rats was significantly reduced and reversed by 1-MT, while KMO expression was not significantly changed (Figure 6C–E).

In the IL, KA levels decreased but were not reversed by the intra-IL injection of 1-MT (Figure 6F). In addition, 3-HK significantly increased, and this increase was reversed by the intra-IL injection of 1-MT (Figure 6G). There was no difference in KAT II expression in the IL of ICV-STZ rats, while KMO expression was significantly increased and reversed by 1-MT (Figure 6H–J).

These results showed that ICV-STZ exerted different effects on the Kyn metabolic branches in the PrL and IL subregions, mainly in the form of attenuated Kyn-KA branches in the PrL and enhanced Kyn-3-HK branches in the IL.

### 2.7. Synaptic Plasticity Was Impaired in Both the PrL and IL in ICV-STZ Rats and Reversed by 1-MT

Abnormal IDO-related Kyn metabolism occurring in glial cells can promote neurogenesis and neuroplasticity [29,30]. In addition, depression is highly correlated with impaired neuroplasticity [31,32]. In the present study, Golgi staining was used to observe dendritic spines, which revealed changes in synaptic plasticity and function (Figure 7A). 

In the PrL, ICV-STZ rats exhibited significant decreases in dendritic spine density and the proportion of mushroom spines, which were reversed by the intra-PrL injection of 1-MT (Figure 7B–D). Additionally, the proportion of thin and filopodia spines in the PrL was significantly increased by ICV-STZ. The intra-PrL injection of 1-MT reversed the increase in thin spines, whereas the increase in filopodia spines was slightly but not significantly blocked by 1-MT (Figure 7D). 

In the IL, ICV-STZ significantly decreased dendritic spine density and the proportion of mushroom spines, which were reversed by the intra-IL injection of 1-MT (Figure 7E–G). In contrast, the proportion of stubby spines in the IL was increased significantly by ICV- STZ, which was reversed by the intra-IL injection of 1-MT (Figure 7G).

Brain-derived neurotrophic factor (BDNF) is a dimeric protein that is often used as an indicator of neuroplasticity. The expression of BDNF in the PrL and IL markedly decreased in ICV-STZ rats, and the intra-PrL and intra-IL administration of 1-MT reversed these ICV-STZ-induced decreases in BDNF levels (Figure 7H–K).

## 3. Discussion

ICV-STZ is a classical model used to simulate sporadic AD. In our previous study, ICV-STZ rats exhibited typical pathological changes similar to sAD in humans, such as the hyperphosphorylation of Tau in the brain, and they progressively showed impairments in learning and cognition beginning on d 14 [33]. In the present study, we found no such pathological changes or learning and memory impairments on d 7 after ICV-STZ administration, but depression-like behaviors appeared. In addition, ICV-STZ rats showed both depression-like behavior and memory impairment on d 14, and their cognitive symptoms further worsened on d 21, but depression-like behaviors disappeared (Figure 1). Forsell et al. [34] indicated that depressive symptoms progressively worsen as AD develops from mild to moderate dementia but become less common with severe dementia. This was consistent with our observations in ICV-STZ rats, suggesting that the ICV-STZ model may be a valid animal model for studying early depression associated with sAD.

The Kyn pathway is the main route for the metabolism of the essential amino acid Trp and is recognized as a major pathway connecting multiple systems, such as inflammation, immune response, neurotransmitter transmission, and oxidative stress [35]. Various enzymes and metabolites of the Kyn pathway are closely associated with neurological and psychiatric disorders [36,37,38]. Tryptophan 2,3-dioxygenase (TDO), IDO1, and IDO2 are the first rate-limiting enzymes in the Kyn pathway. It is generally accepted that TDO, which is predominantly located in the liver, is the main enzyme under physiological conditions, and its expression is regulated mainly by systemic corticosteroid levels [39]. Growing evidence supports that the accelerated production of systemic and central Kyn associated with inflammation is largely dependent on IDO1, whereas the physiological function of IDO2 and its role in disorders involving KP activity is currently unknown [39]. In this study, we found that ICV-STZ activated IDO in some brain regions that are associated with emotion. Among them, the PrL and IL draw our attention since the ratio of Kyn/Trp and IDO activity significantly increased, which implies IDO was activated in these brain regions on d 7 when the rats exhibited depression-like behavior (Figure 2). IDO activation is associated with several neurological disorders, and Kyn pathway abnormalities were also found in different models of AD [18,40] and depression [41,42]. Based on these studies, we hypothesized that the activation of the IDO-related Kyn pathway in the PrL and IL might underlie depression-like behavior in ICV-STZ rats. To confirm this hypothesis, we microinjected 1-MT, an IDO inhibitor, into the PrL or IL to block Kyn metabolism. In several animal models, 1-MT has been confirmed to have antidepressant effects via multiple routes of administration [43,44]. The subcutaneous administration of the IDO inhibitor 1-MT was shown to improve depression-like behaviors in ICV-STZ mice [18]. However, the mechanism of action of 1-MT in the central nervous system, especially in subregions of the mPFC, has not been studied. We found that intra-PrL or intra-IL injections of 1-MT decreased IDO activation and improved multiple depression-like behaviors in ICV-STZ rats (Figure 3). These results suggested that IDO in the PrL or IL may play a key role in the regulation of depression-like behaviors in ICV-STZ rats. Although these findings require deeper investigation, they would reflect region-dependent Trp metabolism balances, also suggesting that AD-related depression may have a unique pathogenesis.

A massive body of evidence suggests that neurological and neurodegenerative disorders emerge from abnormalities in astrocytes and microglia [16,45]. In the central nervous system, glia respond positively to neuroinflammation via the regulation of IDO-related Kyn metabolic balance [46,47]. Homeostatic imbalance in Kyn downstream metabolites was suggested to be regulated by microglia and astrocytes, which might be a prominent mechanism of depression-like behaviors after IDO activation [16]. Although no differences were observed in the structural distribution of glial cells in the PrL and IL [48,49], they appear to play different roles in regulating depression-like behaviors by widespread mechanisms, including synaptic plasticity [50], GABAergic networks [51], and astroglial glutamate transporters [52]. This study showed that astrocytes in the PrL and microglia in the IL were involved in regulating depression-like behaviors in the ICV-STZ model (Figure 4 and Figure 5). GFAP is an important component of the cytoskeleton in astrocytes. A consistent decrease in GFAP expression has been found in the frontal cortex tissue of major depressive disorder patients, and there is evidence that decreased GFAP expression in corticolimbic regions may be a common neurobiological deficit associated with depression [53]. GLAST and GLT-1 are the primary transporters responsible for synaptic glutamate uptake and play an important role in maintaining the physiological functions of astrocytes [54]. In neuropsychiatric disorders and, most notably, depression, reduced levels of GLAST and GLT-1 have been observed in many different brain regions [53,54]. Our results showed a decrease in the number of GFAP-positive cells and the expression of GFAP, GLAST, and GLT-1 proteins in the PrL, indicating deficits in astrocytes and astrocytic dysfunction. Three-dimensional reconstructions of astrocytes also showed a decrease in the surface area and branch complexity of astrocytes in the PrL, suggesting the atrophy and degeneration of astrocytes. These abnormalities in astrocytes were alleviated by the intra-PrL injection of 1-MT. However, in the IL, we did not observe significant abnormalities in astrocytes when ICV-STZ rats exhibited depression-like behaviors. Instead, abnormal activation in microglia might be an adverse consequence of IDO activation in the IL. The decrease in KA in the IL induced by ICV-STZ and the therapeutic effect of 1-MT was not observed in the IL, suggesting that the decreased KA levels in the IL may be attributable to Kyn being more metabolized into 3-HK under ICV-STZ rather than ICV-STZ-related astrocytic deficits. In addition, we also observed differences between these two subregions in the ICV-STZ model in terms of microglia. Iba1 is a microglia/macrophage-specific calcium-binding protein. Recent studies have confirmed that the increase in Iba1 expression in activated microglia is associated with Tau pathology and neuronal apoptosis in AD mouse models [20,55,56]. Resting microglia are characterized by a small soma with multiple protrusions to sense changes in the surrounding environment. When activated, microglia take on a characteristic “amoeba-like” shape, as evidenced by retracted protrusions, enlarged soma, and decreased branches [17]. We found that ICV-STZ activated microglia in the IL, as evidenced by an increase in the number of Iba1-positive cells, Iba1 levels, and pro-inflammatory cytokines (IL-1β and IL-6), together with a concomitant decrease in terminal points and enlargement of the soma volume, all of which were restored by the intra-IL injection of 1-MT. Intriguingly, we did not observe a similar activation of microglia or the therapeutic effect of 1-MT in the PrL. Although we did not observe a decrease in GFAP levels in the IL of ICV-STZ rats, surprisingly, the intra-IL injection of 1-MT significantly increased the expression of GFAP in both the control and model groups, while other astrocyte-related measures did not change significantly. The changes in GFAP induced by 1-MT in IL and its biological mechanism are still unclear and need further study.

In this study, we found that ICV-STZ activated IDO in both the PrL and IL with a different response in glia (Figure 6). Essentially, central KAT II is predominantly localized within astrocytes and contributes to the metabolism of Kyn to KA, while KMO is expressed mainly in microglia and facilitates the conversion of Kyn to 3-HK [57]. KA has been largely described as one of these neuroprotective metabolites capable of antagonizing NMDA receptor-mediated excitatory neurotoxicity and exerting antioxidant effects [58]. Increasing KA levels prevented spatial memory deficits and synaptic loss in mouse models of AD [59]. In contrast with KA, 3-HK is recognized as a neurotoxicogenic factor causing radical-induced oxidative damage, mitochondrial dysfunction, and cell death [60]. AD patients showed high levels of 3-HK in their serum and hippocampi [61,62]. This present study showed that ICV-STZ-induced depression-like behavior was paralleled by a decrease in KA in the PrL and IL, a decrease in KAT II expression in the PrL, and an increase in 3-HK and KMO expression in the IL. Taken together, these results reveal a glia-regional-specific response induced by ICV-STZ and, surprisingly, raise the possibility that although the kynurenine pathway is activated by ICV-STZ in both the PrL and IL, the kynurenine pathway goes in different branches in two subregions. Specifically, the neuroprotective branch is diminished in the PrL, and the neurotoxicogenic branch is enhanced in the IL, which may be involved in the development of ICV-STZ-induced depression-like behaviors. Additionally, the antidepressant effects of 1-MT in the PrL might occur via the normalization of astrocytic deficits and neuroprotective branches, whereas the antidepressant effects of 1-MT in the IL might be at least partially mediated by local anti-inflammatory processes and the blockade of neurotoxicogenic branches.

Human and animal studies indicate that the central inflammatory response may be associated with a reduction in the total volume of a neuron, the shrinkage of dendrites, and the loss of dendritic spines in the mPFC [63,64]. Some studies have shown that under chronic or acute stress, dendritic branches of neurons retract in the PrL [65] and IL [66]. Although glial cells in the PrL and IL regulate depression-like behaviors through different mechanisms, neurons in these two subregions exhibit similar morphological damage, such as reduced synaptic plasticity (Figure 7). In this study, dendritic spine density, the proportion of mushroom-type spines, and BDNF levels in the PrL and IL significantly decreased in ICV-STZ rats. Mushroom-type spines are the main type of spine involved in the regulation of synaptic plasticity because they are more stable and have larger postsynaptic densities and contact areas than other spine types [67]. Notably, these two subregions differed with regard to alterations in the proportions of dendritic spine types in ICV-STZ rats. Increases in thin-type and filopodia-type spine proportions on dendrites were found in the PrL. In the IL, a larger proportion of stubby-type spines was mainly detected. These differences indicated that the deficiency in the neuroprotective effects from astrocytes and the inflammatory stress from microglia induced different dynamic evolution in terms of synaptic plasticity. Although 1-MT selectively restored distinct glial cell abnormalities in the PrL and IL, 1-MT administration significantly improved neuroplasticity and BDNF deficiency in both subregions, suggesting an additional antidepressant mechanism that is implicated in the structural remodeling of dendritic spines.

There is growing evidence indicating that conventional antidepressants cannot effectively ameliorate depressive symptoms in AD patients [68,69], suggesting that depression accompanied by AD may have a different pathogenesis. According to our study, the modulation of the kynurenine pathway may be a more effective treatment strategy for AD-associated depression. Although a clear explanation for the mechanism of sAD formation is still lacking, the ICV-STZ rodent model might make a valuable contribution to the field of sAD model research. It is necessary to further study whether this animal model can simulate clinical processes that are associated with sAD. Although 1-MT injections in both the PrL and IL had antidepressant effects, we observed region-dependent differences in these subregions, suggesting the importance of further investigation on this topic at the therapeutic level. A main issue of the present study was the absence of the measure of quinolinic acid levels because of methodological concerns. Consequently, one of the future developments of the present investigation will be the analysis of this important Kyn metabolite in order to clarify its possible involvement in the observed changes in the ICV-STZ rat model and its potential role in depression signs linked to AD evolution.

## 4. Materials and Methods

### 4.1. Animals

A total of 232 adult male Sprague–Dawley rats (8 ± 1 weeks old, 250–280 g) were procured from the Animal Center of Peking University (Beijing, China). All rats were housed individually in plastic cages with ad libitum access to food and water at an optimum temperature of 25 ± 2 °C and 55–65% relative humidity. We chose the regular maintenance diet for rats, which ensured that they were well nourished. A bottle was positioned in the middle of the cage cover to eliminate location bias. A 12 h/12 h light/dark cycle (lights on at 9:00 a.m.) was regulated in the animal house. All the rats were allowed to acclimate for 7 d before receiving any experimental manipulation. All the experimental procedures complied with the guidelines of the “Animal Research: Reporting of In Vivo Experiments (ARRIVE)” [70] and were carried out in accordance with the National Research Council’s Guide for the Care and Use of Laboratory Animals. The animal experiment protocol was approved by the Peking University Committee on Animal Care and Use (permission no. LA 2020279). During the experiments, rats may have suffered pain from surgery, forced swimming tests, and other experiments. Therefore, we tried to minimize the pain and stress that the rats suffered, and the care process is described in detail in each of the next experimental methods.

### 4.2. Experiment Design

The experimental design is presented in Figure 8A. Experiment 1 was carried out to monitor behavioral changes after STZ injection through a battery of behavioral tests. Experiments 2 and 3 were performed to evaluate molecular and cellular changes in the brain at the time of the onset of depression-like signs. In experiment 1, five rats with surgery-related infections were excluded. Then, 40 rats were randomly grouped into 5 groups (8 animals in each group) as follows: (1) control: no surgery; (2) vehicle: ICV-aCSF; (3) ICV-STZ 7 d; (4) ICV-STZ 14 d; and (5) ICV-STZ 21 d. Behavioral tests, performed as previously described [71,72], were conducted on d 7, d 14, and d 21, respectively. The open field test (OFT) was performed at 9:00 a.m., and the sucrose preference test (SPT) was performed at 12:00 p.m. Following a break, the forced swim test (FST) started at 4:00 p.m. The novel object recognition test (NORT) was performed at 9:00 a.m. on d 8, d 15, and d 22. In experiment 2, 24 rats were randomly grouped into 4 groups (6 animals in each group) for the intra-PrL microinjection experiment as follows: (1) Vehicle (ICV)+Vehicle (PrL); (2) Vehicle (ICV)+1-MT (PrL); (3) STZ (ICV)+Vehicle (PrL); and (4) STZ (ICV)+1-MT (PrL). In experiment 3, 24 rats were randomly grouped into 4 groups (6 animals in each group) for the intra-IL microinjection experiment as follows: (1) Vehicle (ICV)+Vehicle (IL); (2) Vehicle (ICV)+1-MT (IL); (3) STZ (ICV)+Vehicle (IL); and (4) STZ (ICV)+1-MT (IL). After the surgical intervention, animal body weight and food consumption were measured daily. Water intake was measured daily from d 1 to d 5 before food and water deprivation. All behavioral tests were conducted on d 7. The OFT was performed at 9:00 a.m., and the SPT was performed at 12:00 p.m. The novelty-suppressed feeding test (NSFT) was performed at 2:00 p.m. Then, the rats were allowed to eat and drink freely until the FST started at 4:00 p.m. The rats were decapitated on d 8, and the tissue was used for HPLC–MS/MS and Western blot analysis. Since ELISA, IDO activity, immunofluorescence, and Golgi staining require different treatments of tissues, we performed the above biochemical assays on the other three independent experiments identical to experiments 2 and 3.

### 4.3. Surgery for Cannula Implantation

The rats were anesthetized with isoflurane (5% induction and 2% maintenance) and placed in a stereotaxic frame. A unilateral and a bilateral guide cannula (O.D. 0.64 mm × I.D. 0.25 mm, C.C. 1.2 mm, RWD Life Science Co Ltd., Shenzhen, China) were chronically implanted in the lateral ventricle (−0.8 mm AP; −1.5 mm ML; −3.5 mm DV) and mPFC (PrL: +3.2 mm AP; ±0.60 mm ML; −2.6 mm DV, IL: +3.2 mm AP; ±0.6 mm ML; −3.8 mm DV) [73], respectively. The cannula and three anchor screws were bonded to the skull with dental acrylic and dental cement. After surgery, the rats were placed on a heated pad, observed for at least 30 min, and then returned to their original cage. The rats were handled and checked for signs of pain and distress at least once daily. Topical analgesic (lidocaine/prilocaine cream) was used to prevent signs of pain and/or discomfort for at least 2–3 d. The rats were intramuscularly injected with penicillin (400,000 IU/kg) for at least 3 d and allowed to recover for 7 d before the experiments began. The injection placements were verified by the unaided eye or Nissle staining and are shown in Figure 2B and Figure 8A.

### 4.4. Drugs and Treatment

The ICV-STZ procedure was used as previously described [74,75]. After 7 d of recovery from surgery, streptozotocin (Sigma–Aldrich, Cat# S0130, Saint Louis, MO, USA) was administered on d 1 and d 3 at 3:00–5:00 p.m. at a dose of 3 mg/kg dissolved in 4 ± 1 μL artificial cerebrospinal fluid (aCSF; Tocris Bioscience, Cat# 3525/25 ML, Bristol, UK) according to body weight. Streptozotocin was slowly injected into the lateral ventricle using a Hamilton microsyringe fixed to the syringe pump (RWD Life Science Co Ltd., Shenzhen, China) and connected to the injection cannula at a rate of 0.8 μL/min. The needle was left in place for an additional 3 min to allow for diffusion. Vehicle rats were administered with an equal volume of aCSF.

The racemic mixture of 1-MT (1-Methyl-DL-tryptophan; Sigma–Aldrich, Cat# 860646, Saint Louis, MO, USA) was dissolved in 1 M HCl with the pH adjusted to 6.5 using NaOH, and a stock solution at a concentration of 1 mg/mL was prepared with 0.9% sterile sodium chloride solution [76]. Then, it was diluted using 0.9% sterile sodium chloride solution to the final treatment concentration of 50 μg/mL, according to the antidepressant effective dose of 1-MT [77]. From d 1 to d 8, 1-MT was injected once daily into the PrL or IL bilaterally (0.2 μL per side) at a rate of 0.1 μL/min at 9:00–11:00 a.m. The injection cannula was kept in place for another 2 min to allow the drug to diffuse from the tip entirely. Vehicle rats were administered with an equal volume of 0.9% sterile sodium chloride solution.

### 4.5. Nissl Staining

Nissl staining was used to confirm cannula placement and carried out using a Nissl stain Kit (Solarbio, Cat# G1430, Beijing, China). Anesthetized rats were perfused slowly with 200 mL of 0.01 M phosphate-buffered saline (PBS) and 200 mL of 4% paraformaldehyde. Whole brains were immediately removed, soaked in 4% paraformaldehyde at 4 °C for 24 h, and then successively transferred to 20% sucrose and 30% sucrose at 4 °C until tissues were sunk. The brains were rapidly frozen in liquid optimal cutting temperature compound (Sakura Finetek, Cat# 4583, Torrance, CA, USA) and cooled with a mixture of solid carbon dioxide and ethanol. Serial coronal brain sections were cut into 20 μm thickness on a cryostat microtome (Leica Microsystems U.K., Leica CM1950, Milton Keynes, UK) and stored at −20 °C in cryoprotectant solution (48% PBS, 30% ethylene glycol, 20% glycerol, 2% DMSO). Brain sections were stained with Reagent A (Cresyl violet stain) for 1 h at 56 °C. Subsequently, the sections were washed with distilled water, immersed in Reagent B (Nissl Differentiation) for seconds to 2 min, and dehydrated in absolute ethanol. Finally, the sections were cleared in xylene before being cover-slipped with neutral balsam (Solarbio, # G8590, Beijing, China). The image was scanned by a Nikon DS-Ri2 microscope camera (Nikon, Tokyo, Japan) acquired at 1 × magnification. The cannula placements of the lateral ventricle are shown in Figure 8B.

### 4.6. Behavioral Assessment

#### 4.6.1. Forced Swim Test

The FST procedure was conducted according to our previous study [72]. On the pretest day, each rat was individually placed for 15 min into a 25 cm diameter × 60 cm high Plexiglas cylinder filled with 24 ± 1 °C water to a depth of 40 cm. On the test day, the rat was placed into the same cylinder again and recorded for 5 min. The water was changed between testing sessions. Behavior was recorded using two video cameras (one on top and one on the side). After the experiment, the rats were removed from the water, dried with a towel, and returned to their home cage. The videotapes were analyzed by a researcher who was blinded to each rat’s treatment condition. Immobility was defined as the minimum movement necessary to keep the rat’s head above the water. Increased immobility time indicated a state of helplessness.

#### 4.6.2. Sucrose Preference Test

The SPT was used to determine anhedonia-like behavior, which is considered a core symptom of depression. The SPT was performed according to a modified version of the paradigm [78]. During the training phase, rats were habituated to drinking from two bottles of 1% sucrose for 48 h. After training, the rats were deprived of food and water for 24 h before the test. On the test day, one bottle containing 1% sucrose solution in tap water and the other containing tap water alone were placed in the rat’s home cages simultaneously, and rats were allowed to drink freely from both bottles for 1 h. Water and sucrose consumption was measured by comparing the weight difference in the bottles before and after the test. Anhedonia was assessed as sucrose preference, which was calculated according to the following formula: sucrose preference = sucrose intake [g]/(sucrose intake [g] + water intake [g]) × 100%. To exclude the non-specific suppression of drinking, total fluid consumption was calculated as the sum of sucrose intake and water intake.

#### 4.6.3. Open Field Test

The OFT was performed to evaluate locomotor activity and anxiety-like behavior. The rats were submitted individually to a Plexiglas chamber (40 cm × 40 cm × 65 cm) with white light on the top, and behavior was recorded using an automated video tracking system (DigBehv-LM4, Shanghai Jiliang Software Technology, Shanghai, China). The video files were later analyzed using DigBehv analysis software (LAG-4, Shanghai Jiliang Software Technology, Shanghai, China). In 10 min, total distance is often used as an auxiliary indicator in FST to rule out prolonged immobility time due to reduced locomotor activity. Additionally, time in the center (20 cm × 20 cm in the center area) is considered an indicator of anxiety-like behavior. The apparatus was wiped with 75% alcohol between tests to eliminate any smell. 

#### 4.6.4. Novel Object Recognition Test

The NORT is used to evaluate declarative memory and object recognition and is an essential application in the study of cognitive alterations [79]. The test was performed in a Plexiglas box, as described in 2.6.3, and the task procedure consisted of three phases as follows: habituation, training, and test. The rats were habituated to the test box for 20 min during the habituation phase (24 h before the training phase, usually an extra 10 min after the OFT). During the training phase, two identical objects were placed on opposite walls of the testing box, and the animals were allowed to explore for 10 min. Then, we changed one object to a different shape and color in the test phase, and a test session of 10 min was performed after a retention interval of 2 h. The behavior of the subjects in each trial was recorded on video, and the exploration time was scored by an observer blinded to the experimental conditions. Exploration was defined as sniffing, biting, licking, or touching the object with the nose. Turning around or sitting on the object was not considered exploratory behavior. During the test session, a discrimination index (DI) was calculated using the formula (B − A)/(B + A), with B being the time spent exploring the novel object and A being the time spent exploring the familiar object.

#### 4.6.5. Novelty-Suppressed Feeding Test

The NSFT is most widely used to check the efficacy and efficiency of chronic and sub-chronic antidepressant treatments in rodent models, and it was performed as previously described [80]. The test was performed in an open field containing five to six food pellets placed in the middle of the arena. The rats were deprived of food for at least 24 h before the test. After SPT, individual rats were placed in the corner of the round arena (120 cm in diameter) with dim light on the top and allowed to explore it freely for 10 min. The latency of the rat from leaving the corner of the arena to pick up food was recorded. The latency to feed was measured up to 10 min. An increase in latency to feed is considered a measure of anhedonia based on food. After the NSFT, the rats were returned to their home cages and allowed to eat food. Food consumption within 1 h across the groups was recorded to exclude non-specific ingestive behavior.

### 4.7. Estimation of Serotonin and Kynurenine Metabolites

The concentrations of serotonin (5-hydroxytryptamin, 5-HT), Trp, Kyn, 3-HK, and KA were measured using the HPLC–MS/MS method described by Han et al. [81], with slight modifications. Briefly, tissue was transferred into a new EP tube and then mixed with 90 μL of a prechilled (4 °C) methanol (0.1% formic acid) aqueous (8:2, *v*/*v*) mixture and 10 μL of IS (2-Cl-Phe; 1 μg/mL). The mixture was homogenized using an ultrasonic homogenizer followed by centrifugation at 20,000× *g* for 20 min at 4 °C. After centrifugation, the separated supernatant was transferred into a 2 mL auto-sampler, and 10 μL was injected into the system at a flow rate of 0.4 mL/min. The standard curve was prepared using the same procedure as the brain sample. Each data point was from an individual rat.

The HPLC–MS/MS system consisted of a Dionex UltiMate 3000 HPLC system (Thermo, San Jose, CA, USA) and an API 4000Q Trap mass spectrometer (AB SCIEX, Foster City, USA) equipped with an electrospray ionization (ESI) source interface. The optimized mass spectrometric parameters were set as follows: curtain gas, 15 psi; collision gas, 2; ion spray voltage, 5500 V for positive mode or −4500 V for negative mode; ion source temperature, 600 °C; ion source gas 1, 55 psi; and ion source gas 2, 55 psi. Accurate quantification was operated in the multiple reaction monitoring (MRM) mode; the transitions were *m*/*z* 177.1→160.1 for 5-HT (positive), *m*/*z* 205.1→146.1 for Trp (positive), *m*/*z* 209.1→94.1 for Kyn (positive), *m*/*z* 225.1→208.1 for 3-HK (positive), *m*/*z* 188.0→144.0 for KA (negative), *m*/*z* 200.0→154.0 for IS (positive), and *m*/*z* 198.0→181.0 for IS (negative). The declustering potential (DP) was set at 35, 40, 60, 45, and −40 V, and the collision energy (CE) was 12, 24, 22, 14, and −22 V for 5-HT, Trp, Kyn, 3-HK, and KA, respectively. Under the condition of the initial mobile phase and flow rate, the working pressure values were around 170 bar (about 2466 psi), and the working pressure values gradually decreased as the proportion of acetonitrile in the mobile phase increased.

Chromatographic separation was performed on an Ultimate XB-C18 column (100 mm × 2.1 mm, 5 μm, Welch Materials, Inc., Shanghai, China). Mobile phase A was water containing 0.1% formic acid, and mobile phase B was acetonitrile. The temperature of the auto-sampler was 4 °C. Gradient separation was set as follows: 0–1 min, 5% B; 1–3 min, 5–60% B; and 3.1–5 min, 5% B for column equilibration. The analysis was performed for a total run time of 5 min. Under these conditions, the retention times were 1.40, 3.53, 1.95, 0.89, 3.68, 3.49, and 3.47 min for 5-HT, Trp, Kyn, 3-HK, KA, IS (positive), and IS (negative), respectively. Detailed parameters, linearity, accuracy, precision and typical chromatograms are shown in Appendix A. The above data were recorded and analyzed using AB SCIEX Analyst 1.6 software.

5-hydroxytryptamin (5-HT, purity ≥ 98%, Cat# H9523), Tryptophan (Trp, purity ≥ 99.5%, Cat# 93659), kynurenine (Kyn, purity ≥ 98%, Cat# K8625), 3-hydroxy-DL-kynurenine (3-HK, purity ≥ 98%, Cat# 148776), and kynurenic acid (KA, purity ≥ 98%, Cat# K3375) were purchased from Sigma–Aldrich (Saint Louis, MO, USA). 2-Chloro-L-phenylalanine (2-Cl-Phe, purity = 98%, Cat# C105993), used as an internal standard (IS), was purchased from Aladdin Inc. (Shanghai, China). HPLC-grade acetonitrile and methanol were obtained from Fisher Chemical (Fisher Scientific, Shanghai, China).

### 4.8. Evaluation of Indoleamine 2,3-Dioxygenase (IDO) Enzyme Activity

The quantification of IDO activity was measured using an indoleamine 2,3-Dioxygenase 1 (IDO1) Activity Assay Kit (Sigma–Aldrich, Cat# MAK356, Saint Louis, MO, USA). The tissue was homogenized using a Dounce homogenizer in an ice-cold IDO1 assay buffer with a protease inhibitor cocktail containing PMSF. The mixture was then incubated on ice for 5 min and centrifuged at 10,000× *g* for 15 min at 4 °C. Subsequently, the supernatant was collected, and the reagents or samples were added to a 96-well plate as directed and incubated at 37 °C for 45 min in the dark. Then, 50 μL of Fluorescent Developer Solution was added to each well and incubated for 3 h at 45 °C in the dark with gentle shaking. Finally, the plates were cooled to room temperature for 1 h, and fluorescence was measured in end-point mode (λex/nm = 402/λem/nm = 488).

### 4.9. Immunofluorescence Staining and Image Analysis

Immunofluorescence staining was performed as described previously [82]. The steps for obtaining brain sections were described in detail in 4.4. To label the astrocytes and microglia, a multiple-color immunochemistry kit (Absin, Cat# abs50029, Shanghai, China) was used following the manufacturer’s instructions. Briefly, the sections were first placed in PBS (3 × 5 min) to wash out the cryoprotectant solution. Then, they were incubated in cold acetone for 5 min, followed by washing in PBS (3 × 5 min). Antigen retrieval was conducted in antibody eluent (Absin, Cat# abs994, Shanghai, China) for 40 min at 37 °C. After washing in PBS (3 × 5 min), the sections were permeabilized with 0.3% Triton X-100 (Sigma–Aldrich, Cat# T9284, Saint Louis, MO, USA) for 20 min and blocked with 5% donkey serum at room temperature for 40 min. The sections were then incubated with primary antibodies against GFAP (1:500; rabbit mAb, Cell Signaling Technology, Cat# 80788, Danvers, MA, USA) for 1 h, and rabbit HRP-conjugated secondary antibody was applied and incubated for 1 h. Next, the sections were washed in TBST (3 × 5 min) and incubated with Tyramide Signal Amplification (TSA) reagent for 10 min. The above operation was repeated, followed by incubation with primary antibodies against Iba-1 (1:500; Rabbit mAb, Cell Signaling Technology, Cat# 17198, Danvers, MA, USA). Finally, the nuclei were subsequently stained with DAPI.

Single images from the PrL and IL for each rat were scanned using a Leica TCS-SP8 STED 3X laser scanning confocal microscope (Leica, Wetzlar, Germany). Z stacks were performed with 1 μm steps in the z-direction and recorded with 1024 × 1024-pixel resolution in the x–y direction. Three-dimensional reconstructions of astrocytes and microglia were analyzed using Imaris software Version 9.0.0 (Bitplane, Concord, MA, USA). Morphological changes in microglia were evaluated following a previously published protocol [83] with modification. Briefly, a new surface was created, and the parameters were set to smooth = 0.1, threshold = 50, and filter (number of voxels) = 20. In edit mode, 8-10 glial cells with regular and clear shapes were selected in the whole-image field. Then, the surface area and Sholl intersection number of GFAP^+^ cells and terminal points and soma volume of Iba1^+^ cells were examined in the statistics tab. Sholl analysis was carried out by drawing concentric circles as a step by 5 μm, and the number of intersections between the circles and microglia branches was counted, as shown in Appendix A. These morphological parameters of astrocytes and microglia can reflect complexity and distinguish the state of the glia.

### 4.10. Western Blot

After the rats were decapitated, the brains were immediately removed to a prechilled brain matrix with a 1.0 mm coronal slice thickness (RWD Life Technology, Shenzhen, China). PrL and IL tissues were harvested on ice guided by the Paxinos and Watson rat brain atlas [73] and stored separately in prechilled microcentrifuge tubes at −80 °C until assayed. The tissue was homogenized in RIPA buffer (Solarbio, Cat# R0010, Beijing, China) supplemented with protease inhibitors (Solarbio, Cat# P0100, Beijing, China) and phosphatase inhibitors (Solarbio, Cat# P1260, Beijing, China). Protein (45 μg) was separated via 10% sodium dodecyl sulfate–polyacrylamide gel electrophoresis. ColorMixed Protein Marker 180 (ABclonal, Cat# RM19001, Wuhan, China) was used for monitoring protein separation during SDS–polyacrylamide gel electrophoresis, verification of Western transfer efficiency on membranes, and approximating the size of the proteins. This maker was a ready-to-use three-color protein standard with 10 pre-stained proteins covering a wide range of molecular weights from 10 to 180 kDa. After electrophoresis, the gels were cut according to the markers. The gel loaded with proteins from mPFC tissue was cut at 75 kDa, 45 kDa, and 25 kDa. The 75 kDa–45 kDa section was used for detecting Aβ, Tau, and p-Tau^Ser199^, and the 45 kDa–25 kDa section was used for detecting β-actin. Three different gels were prepared for PrL or IL tissue. The first gel was cut at 75 kDa, 45 kDa, and 25 kDa. The 75 kDa–45 kDa section was used for detecting GFAP, GLAST, and GLT-1; the 45 kDa–25 kDa section was used for detecting β-actin; and a 25 kDa–10 kDa section was used for detecting BDNF. The second gel was cut at 60 kDa and 35 kDa. The 60 kDa–35 kDa section was used for detecting β-actin, and the 35 kDa–10 kDa section was used for detecting Iba 1. The third gel was cut at 75 kDa, 45 kDa, and 25 kDa. The 75 kDa–45 kDa section was used for detecting KAT II and KMO, and the 45 kDa–25 kDa section was used for detecting GAPDH and IDO. The gels with high-molecular-weight proteins were transferred to polyvinylidene fluoride (PVDF) membranes (0.45 μm; Millipore, Boston, MA, USA) for 2 h at a constant current of 200 mA. Additionally, the gels with small-molecular-weight target proteins (BDNF and Iba1) were transferred to PVDF membranes for 0.5 h. Then, the membranes were blocked with 5% skim milk for 1 h at room temperature and incubated with primary antibodies, including anti-GAPDH (1:10,000; Rabbit pAb, ABclonal, Cat# AC001, Wuhan, China), anti-β-actin (1:4000; Rabbit mAb, ABclonal, Cat# AC038, Wuhan, China), anti-β-Amyloid (Aβ) (1:50; Mouse mAb, Santa Cruz Biotechnology, Cat# sc-28365, Dallas, TX, USA), anti-Tau (1:1000; Mouse mAb, Cell Signaling Technology, Cat# 4019, Danvers, MA, USA), anti-Phospho-Tau (Ser199) (1:1000; Mouse mAb, Cell Signaling Technology, Cat# 29957, Danvers, MA, USA), anti-IDO (1:1000; Rabbit mAb, Cell Signaling Technology, Cat# 86630, Danvers, MA, USA), anti-KAT II (1:1000; Mouse pAb, Abcam, Cat# ab89608, Cambridge, UK), anti-KMO (1:1000; Rabbit mAb, Abcam, Cat# ab233529, Cambridge, UK), anti-GFAP (1:1000; Rabbit mAb, Cell Signaling Technology, Cat# 80788, Danvers, MA, USA), anti-GLAST (EAAT-1 in humans, 1:1000; Rabbit mAb, Cell Signaling Technology, Cat# 5684, Danvers, MA, USA), anti-GLT-1 (EAAT-2 in human, 1:500; Rabbit pAb, Abcam, Cat# ab41621, Cambridge, UK), anti-Iba1 (1:1000; Rabbit mAb, Cell Signaling Technology, Cat# 17198, Danvers, MA, USA), and anti-BDNF (1:500; Rabbit mAb, Abcam, Cat# ab108319, Cambridge, UK) in TBST buffer (Tris-buffered saline +0.1% Tween-20) overnight at 4 °C. The blots were then washed with TBST three times before incubation with HRP goat anti-mouse IgG (H+L) antibody (1:2000; ABclonal, Cat# AS003, Wuhan, China) or HRP goat anti-rabbit IgG (H+L) antibody (1:2000; ABclonal, Cat# AS014, Wuhan, China) for 2 h at room temperature. After 3 × 5 min TBST washes, an ECL Enhanced Kit (ABclonal, Cat# RM00021, Wuhan, China) was used for detection enhancement, and blots were visualized using ImageJ software (version 1.8.0; NIH, Bethesda, MD, USA). Each data point was from an individual rat, and each group consisted of six or eight rats. The results were normalized to the protein expression level of β-actin or GAPDH. Some target proteins with similar molecular weights were not separated via physical segmentation and transferred to the same PVDF membrane. These PVDF membranes were washed with Western Blot Stripping Buffer (Biotides, Cat# WB1501, Beijing, China) for 25 min to remove the primary antibodies. Then, the membranes were washed several times, blocked, incubated with a secondary antibody, and then re-incubated with ECL Enhanced Kit to confirm the stripping quality. If the primary antibody was effectively removed by the stripping procedure, no secondary antibody should bind to the membrane and no signal should be produced. Subsequently, blocking, primary antibody incubation, secondary antibody incubation, and chemical imaging proceeded for the quantitative analysis of another target protein on the same membrane. These membranes were stripped no more than twice and detected at most 3 different proteins with similar molecular weights.

### 4.11. Enzyme-Linked Immunosorbent Assay (ELISA)

Rat ELISA kits were used to measure the levels of IL-1β (ABclonal, Cat# RK00009, Wuhan, China), IL-6 (ABclonal, Cat# RK00020, Wuhan, China), and TNF-α (ABclonal, Cat# RK00029, Wuhan, China) in the PrL and IL. Briefly, the tissue was homogenized and centrifuged at 2000× *g* for 20 min at 4 °C; then, the supernatant was extracted and the total protein in the samples was quantified by a BCA kit (Thermo, Cat# 23227, Waltham, MA, USA). Another 100 µL of supernatant was incubated in a 96-well plate. The cytokine levels were estimated by interpolation from a standard curve by colorimetric measurements at 450 nm (correction wavelength 630 nm) on an ELISA plate reader (Thermo Fisher Scientific Multiskan Mk3, Waltham, MA, USA). Each data point was from an individual rat. The results are shown as pg/mg of protein.

### 4.12. Golgi Staining

The FD Rapid Golgi-Staining Kit (FD NeuroTechnologies, Cat# PK401, Columbia, MD, USA) was used to reveal the density of the dendritic spines in pyramidal neurons in the PrL and IL. Brains were immersed in Golgi-staining impregnation solution for 2 weeks. Sections were cut at 150 µm on a cryostat at −20 °C to −22 °C. The spines on secondary or tertiary dendrites of pyramidal neurons were calculated at a dendritic segment length of approximately 50 µm. At least 3 dendrites per rat were traced, and a total of 6 rats per group were counted. The images were captured under a Nikon Eclipse Ci-L microscope (Nikon, Tokyo, Japan) using DP controller software (version 3.1.1; Olympus, Tokyo, Japan) with a 100× A/1.25 oil immersion lens. The density of dendritic spines and the dendritic spine morphologies in the PrL and IL were analyzed using ImageJ software (version 1.8.0; NIH, Bethesda, MD, USA). The dendritic spine density was calculated as the total number of spines per 10 µm length of branch. Dendritic spine morphologies were classified into 4 main types as follows: thin, filopodia, mushroom, and stubby [84]. The thin type has a narrow neck with elongated protrusion; the filopodia type is identified as long, thin structures; the mushroom type has a large irregular head with a neck diameter smaller than the head diameter; and the stubby type has no obvious constriction between the protrusion and attachment to the neck. The proportion of each type was quantified as ([spine number with each type/total spine number] × 100).

### 4.13. Data and Statistical Analysis

The sample size was the number of independent values, and statistical analysis was performed using these independent values. The sample sizes are indicated in the figure legends. The data from Western blot were normalized to control group values and determined as “fold change” in the figures. Statistical analysis of the results was carried out with the help of GraphPad Prism software (version 8; GraphPad Software, Inc., San Diego, CA, USA). Statistical analysis was undertaken only for studies where each group size was at least n = 6. The data are presented as means ± SEMs. All statistical tests were two-tailed. An unpaired *t*-test was used to test Western blot data and HPLC-MS/MS for vehicle and ICV-STZ 7 d rats variance. One-way and two-way analysis of variance (ANOVA) followed by Tukey’s multiple comparison test was applied for comparisons among several groups. For two-way ANOVA, the procedure (vehicle or ICV-STZ) and the treatment (saline or 1-MT) were taken as between-group factors. Means ± SEMs of all data were presented in Appendix A. The types of comparison tests involved in each experiment were labeled in the figure legends. *p* < 0.05 was considered statistically significant in all tests. If the results were not significant, only the statistical method used was reported. The experimenter or observer was blind to the groups while analyzing the results.

## 5. Conclusions

Overall, ICV-STZ induced depression-like behavior before cognitive impairment in rats. In ICV-STZ rats, IDO was activated in the PrL and IL, but the mechanisms that were involved in regulating depressive-like behaviors in these two subregions were different. The results provide clear evidence that exploring subregional mechanisms of the mPFC is important for further understanding the pathogenesis and treatment of depression. The selective inhibition of IDO reversed ICV-STZ-induced depression-like behaviors by ameliorating neuroprotective branch deficits in the PrL and inhibiting neurotoxicogenic branch overactivation in the IL. These findings provide valuable information for further understanding the pathogenesis of depression, which may contribute to discovering novel treatment targets for depression, especially among the population that has a high risk of AD.

## Figures and Tables

**Figure 1 ijms-25-07496-f001:**
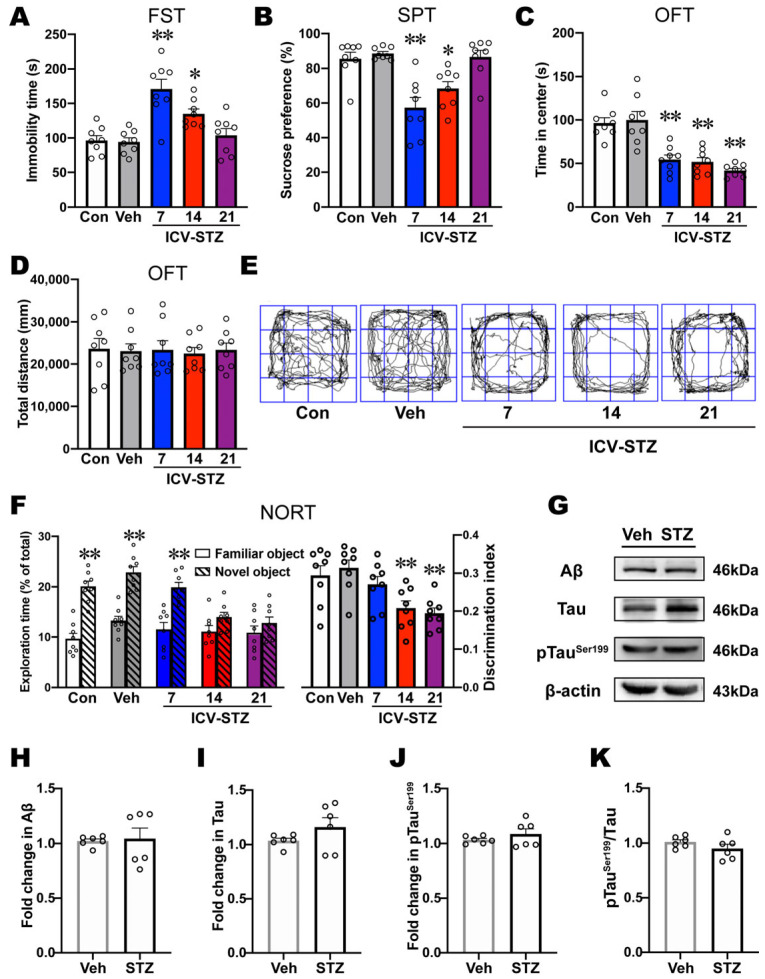
ICV-STZ induced depression-like behaviors before cognitive impairment. (**A**) Immobility time in the FST and (**B**) sucrose preference in the SPT were used to evaluate depression-like behaviors. (**C**) The time spent in the center of the OFT was used to evaluate anxiety-like behavior. (**D**) The total distance traveled in the OFT was used to detect locomotor activity. (**E**) Footprint pattern in the OFT. (**F**) Exploration time and the discrimination ratio in the NORT were used to detect recognition memory and discrimination ability (*n* = 8). (**G**–**K**) Western blots and quantification of Aβ, Tau, pTau^Ser199^ protein levels, and the pTau^Ser199^/Tau ratio in the mPFC. β-actin was used as the quantitative loading control (*n* = 6). The data are expressed as individual values with means ± SEMs. * *p* < 0.05, ** *p* < 0.01 vs. vehicle group; (**A**–**F**), one-way ANOVA followed by Tukey’s multiple-comparison post hoc test; (**H**–**K**), unpaired Student’s *t*-test.

**Figure 2 ijms-25-07496-f002:**
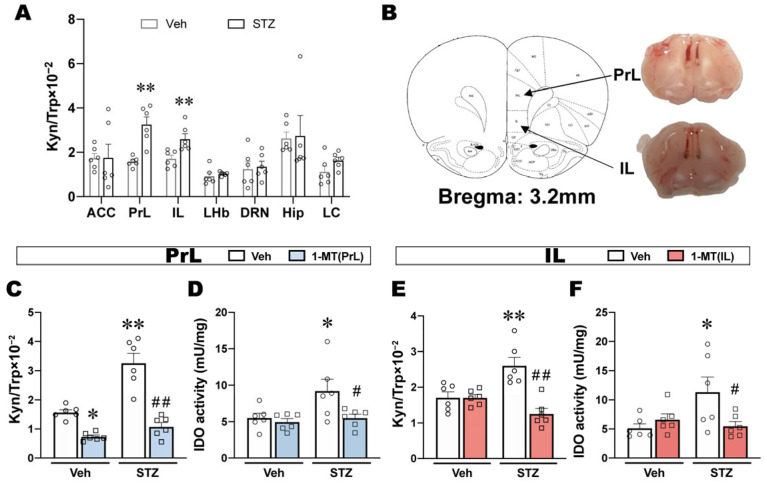
ICV-STZ induced IDO activation in the PrL and IL. (**A**) The Kyn/Trp ratio in several brain regions of ICV-STZ depressed rats (*n* = 6). (**B**) Location of the PrL and IL injection sites. (**C**,**D**) The Kyn/Trp ratio and IDO activity in the PrL (*n* = 6). (**E**,**F**) The Kyn/Trp ratio and IDO activity in the IL (*n* = 6). The data are expressed as individual values with means ± SEMs. * *p* < 0.05, ** *p* < 0.01 vs. Vehicle + Vehicle group; # *p* < 0.05, ## *p* < 0.01 vs. ICV-STZ + Vehicle group; (**A**), unpaired Student’s *t*-test; (**C**–**F**), two-way ANOVA followed by Tukey’s multiple-comparison post hoc test.

**Figure 3 ijms-25-07496-f003:**
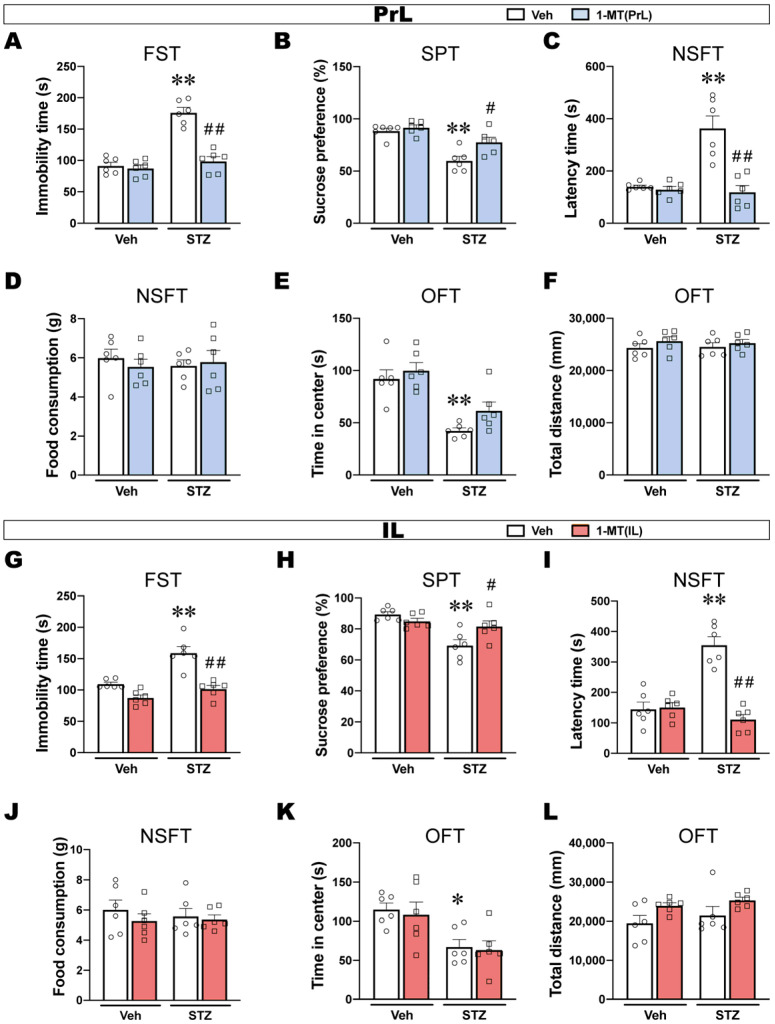
Intra-PrL or IL administration of 1-MT prevented depression-like behaviors induced by ICV-STZ. (**A**,**G**) Immobility time in the FST. (**B**,**H**) Sucrose preference in the SPT. (**C**,**I**) Latency time in the NSFT. (**D**,**J**) Food consumption in the NSFT. (**E**,**K**) Time spent in the center of the OFT. (**F**,**L**) Total distance traveled in the OFT (*n* = 6). The data are expressed as individual values with means ± SEMs. * *p* < 0.05, ** *p* < 0.01 vs. Vehicle + Vehicle group; # *p* < 0.05, ## *p* < 0.01 vs. ICV-STZ + Vehicle group; two-way ANOVA followed by Tukey’s multiple-comparison post hoc test.

**Figure 4 ijms-25-07496-f004:**
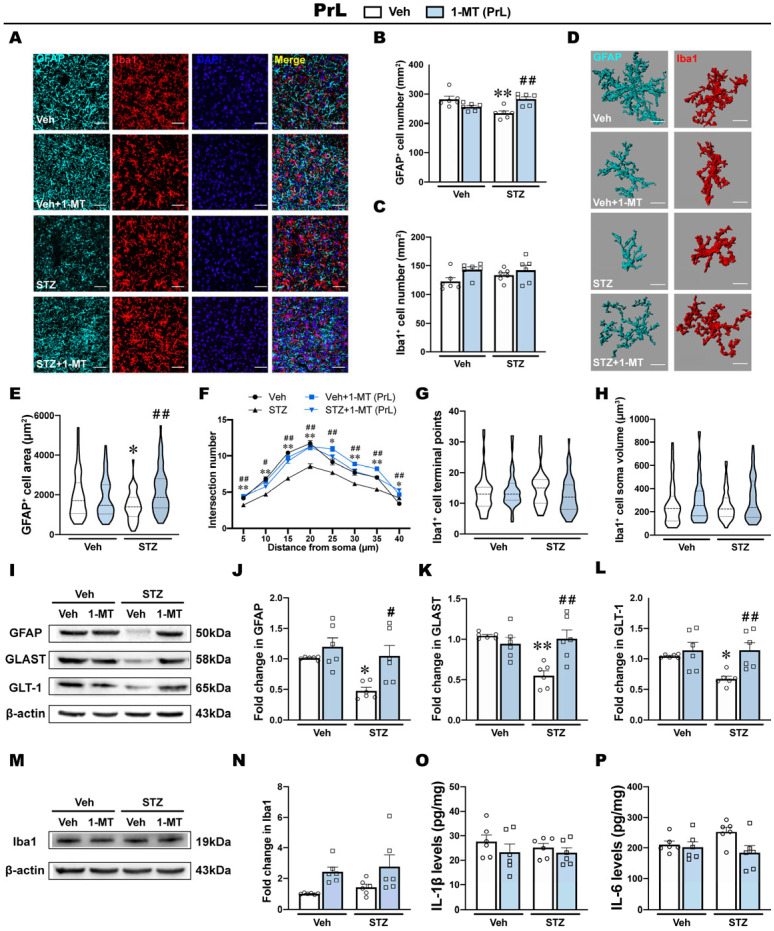
ICV-STZ induced astrocyte defects in the PrL, which were reversed by intra-PrL administration of 1-MT. (**A**) Fluorescent images showing the immunostaining of GFAP and Iba1. GFAP was labeled with TSA monochrome fluorescent dye 570 (cyan). Iba1 was labeled with TSA monochrome fluorescent dye 700 (red). The cell nucleus was counterstained with DAPI (blue). Scale bars = 50 μm. (**B**) Counts of GFAP^+^ cell were used to evaluate the expression of astrocytes in the PrL (*n* = 6 slides from 6 rats). (**C**) Counts of Iba1^+^ cell were used to evaluate the expression of microglia in the PrL (*n* = 6 slides from 6 rats). (**D**) Three-dimensional reconstruction of astrocytes and microglia. Scale bars = 15 μm. (**E**) The GFAP^+^ cell area was used to evaluate morphological alterations in the PrL. (**F**) The line plot of Sholl analysis revealed the number of intersections per 5 μm of astrocytes in the PrL. *n*_1_ = 58, *n*_2_ = 61, *n*_3_ = 52, and *n*_4_ = 65. (**G**,**H**) The number of Iba1^+^ cell terminal points and soma volume were used to evaluate the state of microglia in the PrL. *n*_1_ = 60, *n*_2_ = 57, *n*_3_ = 59, and *n*_4_ = 61. (**I**–**L**) Western blots and quantification of GFAP, GLAST, and GLT-1 in the PrL. (**M**,**N**) Western blots and quantification of Iba1 in the PrL. β-actin was used as a quantitative loading control (*n* = 6). (**O**,**P**) Levels of IL-1β and IL-6 determined via ELISA in the PrL (*n* = 6). The data are expressed as individual values with means ± SEMs. * *p* < 0.05, ** *p* < 0.01 vs. Vehicle + Vehicle group; # *p* < 0.05, ## *p* < 0.01 vs. ICV-STZ + Vehicle group; two-way ANOVA followed by Tukey’s multiple-comparison post hoc test.

**Figure 5 ijms-25-07496-f005:**
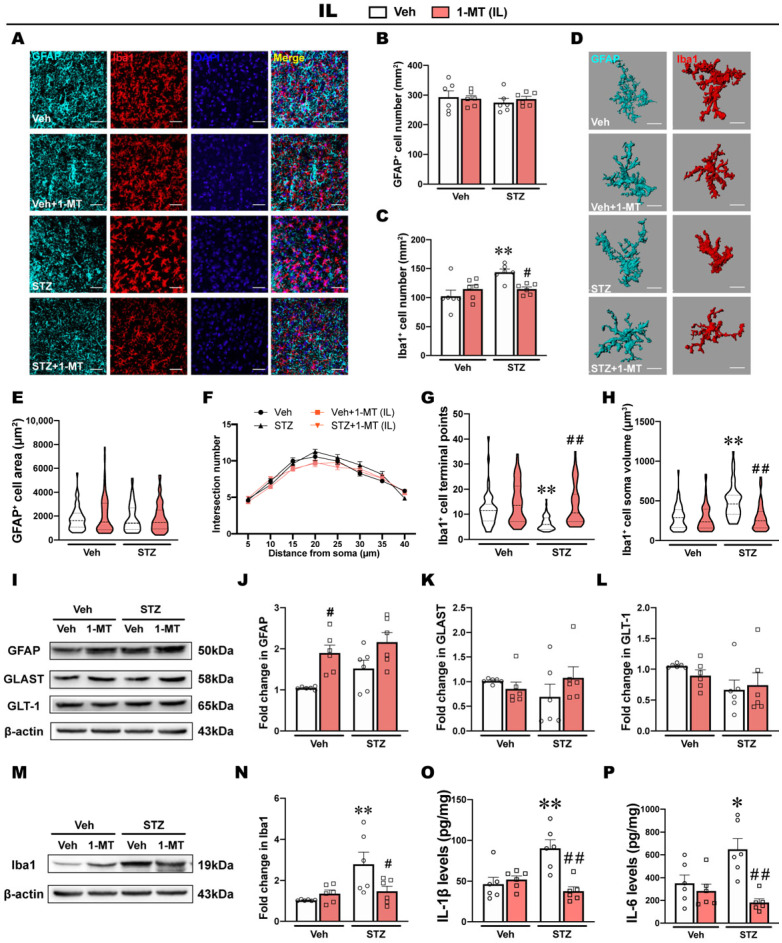
ICV-STZ induced microglia overactivation in the IL, which was reversed via the intra-IL administration of 1-MT. (**A**) Fluorescent images showing the immunostaining of GFAP and Iba1. GFAP was labeled with TSA monochrome fluorescent dye 570 (cyan). Iba1 was labeled with TSA monochrome fluorescent dye 700 (red). The cell nucleus was counterstained with DAPI (blue). Scale bars = 50 μm. (**B**) Counts of GFAP^+^ cell were used to evaluate the expression of astrocytes in the IL (*n* = 6 slides from 6 rats). (**C**) Counts of Iba1^+^ cell were used to evaluate the expression of microglia in the IL (*n* = 6 slides from 6 rats). (**D**) Three-dimensional reconstruction of astrocytes and microglia. Scale bars =15 μm. (**E**) The GFAP^+^ cell area was used to evaluate morphological alterations in the IL. (**F**) The line plot of Sholl analysis revealed the number of intersections per 5 μm of astrocytes in the IL. *n*_1_ = 65, *n*_2_ = 64, *n*_3_ = 57, and *n*_4_ = 58. (**G**,**H**) The number of Iba1^+^ cell terminal points and soma volume were used to evaluate the state of microglia in the IL. *n*_1_ = 48, *n*_2_ = 54, *n*_3_ = 61, and *n*_4_ = 48. (**I**–**L**) Western blots and quantification of GFAP, GLAST, and GLT-1 in the IL. (**M**,**N**) Western blots and quantification of Iba1 in the IL. β-actin was used as a quantitative loading control (*n* = 6). (**O**,**P**) Levels of IL-1β and IL-6 determined via ELISA in the IL (*n* = 6). The data are expressed as individual values with means ± SEMs. * *p* < 0.05, ** *p* < 0.01 vs. Vehicle + Vehicle group; # *p* < 0.05, ## *p* < 0.01 vs. ICV-STZ + Vehicle group; two-way ANOVA followed by Tukey’s multiple-comparison post hoc test.

**Figure 6 ijms-25-07496-f006:**
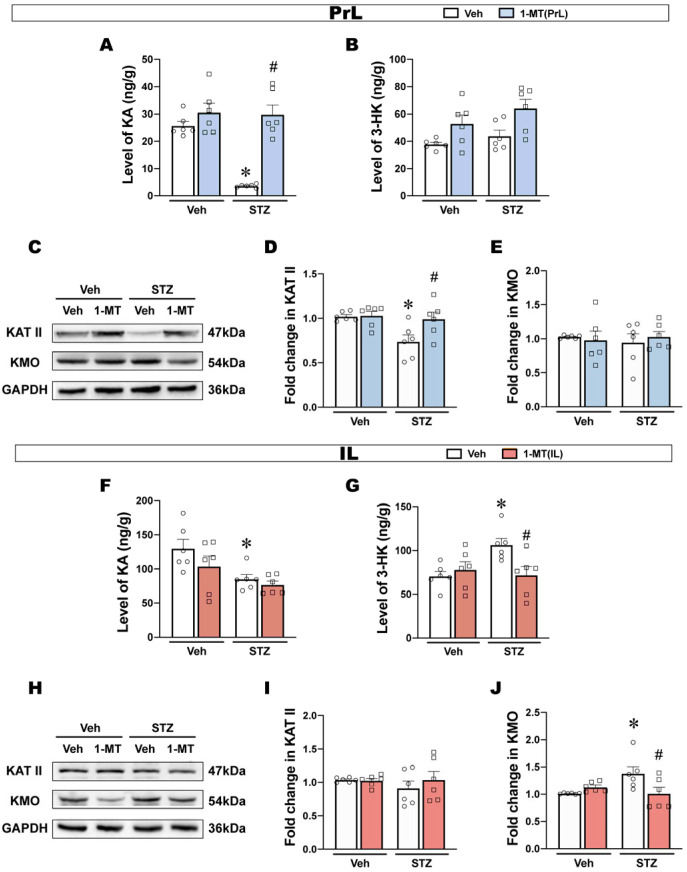
ICV-STZ has different effects on the Kyn pathway branches in the PrL and IL. (**A**,**B**) Levels of KA and 3-HK in the PrL (*n* = 6). (**C**–**E**) Western blots and quantification of KAT II and KMO in the PrL. β-actin was used as a quantitative loading control (*n* = 6). (**F**,**G**) Levels of KA and 3-HK in the IL (*n* = 6). (**H**–**J**) Western blots and quantification of KAT II and KMO in the IL. β-actin was used as a quantitative loading control (*n* = 6). The data are expressed as individual values with means ± SEMs. * *p* < 0.05 vs. Vehicle + Vehicle group; # *p* < 0.05 vs. ICV-STZ + Vehicle group; two-way ANOVA followed by Tukey’s multiple-comparison post hoc test.

**Figure 7 ijms-25-07496-f007:**
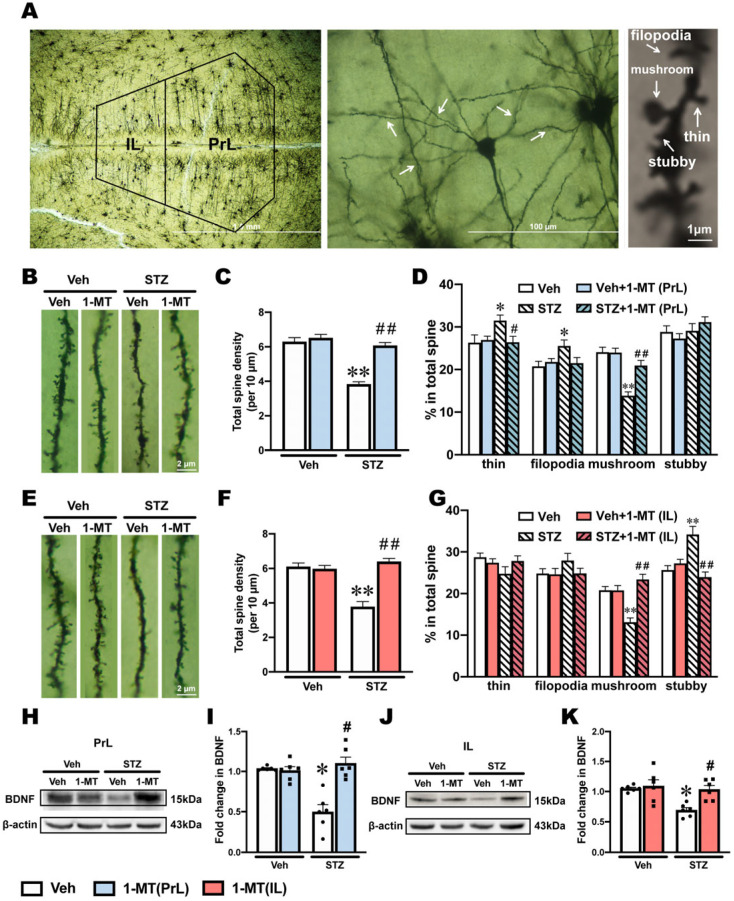
Intra-PrL and -IL administration of 1-MT improved synaptic deficits in the PrL and IL that were induced by ICV-STZ. (**A**) Golgi staining of PrL and IL pyramidal neurons. (**B**) Representative images of dendritic spines in the PrL. (**C**,**D**) The total dendritic spine number and the proportion of each spine type were used to evaluate spine morphology alterations in the PrL (*n* = 18 dendrites from 6 rats in each group). (**E**) Representative images of dendritic spines in the IL. (**F**,**G**) The total dendritic spine number and the proportion of each spine type were used to evaluate spine morphology alterations in the IL (*n* = 18 dendrites from 6 rats in each group). (**H**,**I**) Western blots and quantification of BDNF in the PrL. (**J**,**K**) Western blots and quantification of BDNF in the IL. β-actin was used as a quantitative loading control (*n* = 6). The data are expressed as individual values with means ± SEMs. * *p* < 0.05, ** *p* < 0.01 vs. Vehicle + Vehicle group; # *p* < 0.05, ## *p* < 0.01 vs. ICV-STZ + Vehicle group; two-way ANOVA followed by Tukey’s multiple-comparison post hoc test.

**Figure 8 ijms-25-07496-f008:**
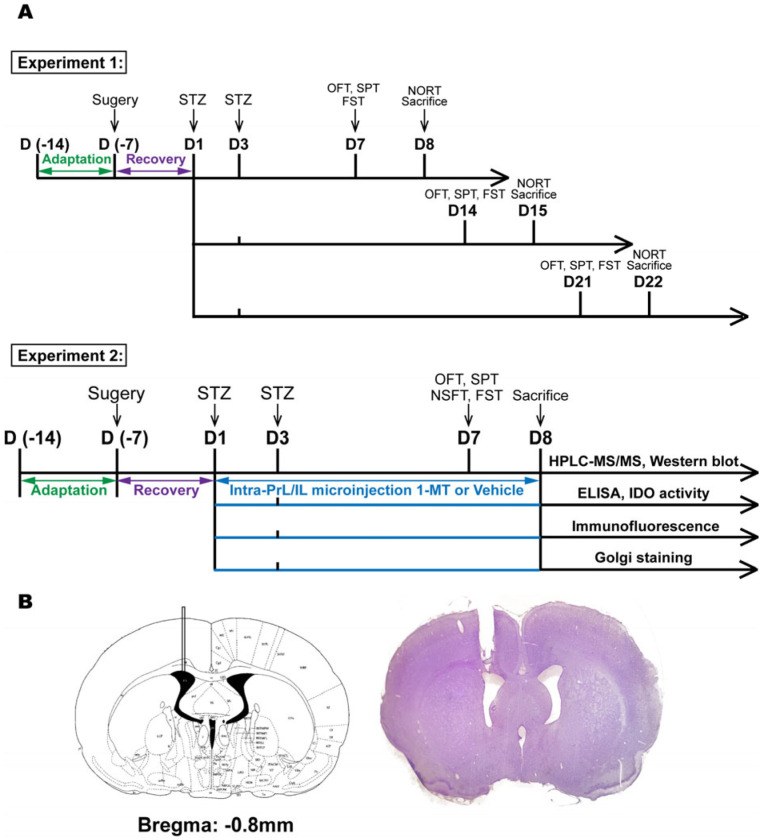
(**A**) Experimental design and (**B**) location of intracerebroventricular injection sites.

## Data Availability

The dataset is available upon request from the authors.

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
