# Peer review of "Inhibition of Indoleamine 2,3-Dioxygenase Exerts Antidepressant-like Effects through Distinct Pathways in Prelimbic and Infralimbic Cortices in Rats under Intracerebroventricular Injection with Streptozotocin"

_ijms, 2024, doi:10.3390/ijms25137496_

Round 1
Reviewer 1 Report (New Reviewer)
Comments and Suggestions for Authors
The study by Qin et al. entitled: “Inhibition of indoleamine 2,3-dioxygenase in the prelimbic and infralimbic cortices exerts antidepressant effect through different pathways in the ICV-STZ rats” reports the involvement of the main enzyme of the kynurenine shunt, indoleamine 2,3-dioxygenase (IDO), in the development of depression-like behaviors occurring after 7 days of intracerebroventricular streptozotocin microinjection (ICV-STZ) in rats, signs preceding the onset of the cognitive phase of Alzheimer’s Disease. The ICV-STZ procedure in rodents induces indeed a condition considered a model of Alzheimer’s Disease.
In particular, ICV-STZ in rodents produces a model of sporadic, non-familial Alzheimer's disease. This model can be used to study and understand the development, evolution, and pathogenetic bases of the major stages of the disease, starting from the early phases, overall defined by psychiatric symptoms and depressive-like behaviors, to the terminal onset of devastating cognitive and neurological deficits. In more detail, the authors used this animal model to study the effects of STZ, at the histological, molecular and behavioral levels, in two different brain areas, the pre limbic and limbic ones. The authors found that STZ is able to induce distinct molecular and cellular alterations in these two brain regions related to the depressive stage arising after 7 days of treatment. In both areas and regardless of the area-specific alterations caused, symptoms were reversed by the IDO inhibitor 1-methyl-tryptophan (1-MT).
The study encompasses many experimental sections and many results have been obtained. Anyway, there are concerns as regards the presentation of this manuscript requiring to be considered in order to attain a better understanding of the investigation aims and result impact. Also, there are issues linked to the experimental design that must be considered before publication.
General comments
Title: I suggest to improve the title as follows: “Inhibition of indoleamine 2,3-dioxygenase in the prelimbic and infralimbic cortices exerts antidepressant-like effects through different pathways in rats under intracerebroventricular injection with streptozotocin”
Abbreviations: Due to the high number of abbreviations used in the manuscript and for a better understanding of the text, I suggest the authors to create an Abbreviations sections at the beginning or at the end of the manuscript, on the bases of the IJMS guidelines.
Specific comments:
Abstract : Abbreviations “Iba-1 positive” and “1-MT” should be first presented as full names with abbreviations in parenthesis.
Introduction
This section is rather confusing and not extensive enough compared to the whole manuscript and results. There is also a sentence that should be rather reported in the results and discussion sections, not in the introduction one. The main scope of the work should be presented more precisely. These are the main points to revise:
- Alzheimer’s disease should be better presented;
- Depression symptoms in Alzheimer’s disease should be better treated, possibly considering their features in respect to non-neurological depression (see, among others, these articles: Aguera-Ortiz et al. Depression in Alzheimer's Disease: A Delphi Consensus on Etiology, Risk Factors, and Clinical Management. Front. Psychiatry, Sec. Aging 2021; 12:1-16; Modrego PJ, de Cerio LD, Lobo A. The Interface between Depression and Alzheimer's Disease. A Comprehensive Approach. Ann Indian Acad Neurol. 2023;26(4):315-325).
- Page 1 line 38: sporadic AD (sAD) – present this AD form through a short sentence;
- Page 1 line 40: Remove “Recently”;
- Page 2 lines 47-49: please change as: “ Furthermore, ICV-STZ also reproduces the AD progression, promoting first non-cognitive features such as depression-like behaviors in rodents”
- Page 2 lines 68-72: As aforementioned, this part should be removed from the Introduction section, since no results should be presented here. The authors should insert it in the Result and Discussion sections.
Overall, the authors should revise their main scopes in this section, especially focusing on the importance to unravel the pathogenesis of the passages from the depressive and non-cognitive stage of AD to the cognitive and memory decreasing one. The ICV technique should be also introduced for readers and scientists less informed about this methodology applied to animals. The behavioral tests employed should be also shortly introduced, as well as the authors should more precisely introduce to the biological and cellular markers used. Essentially, authors have to explain more in detail the study rationale and their model.
Materials and Methods
Animals: Please indicate the animals’ age. Describe food type and animal nutritional conditions. It is also important to report the total number of rats included into the study, which, due to the invasive procedure employed, must be as small as possible. The authors should also report all efforts and procedures carried out to contain any possible animal suffering.
Drugs and treatments: page 17 line 500: what was the starting concentration of 1-methyl-tryptophan ? Was 1-MT a racemic mixture or the L-isomer? Please report these points. Please report also the final 1-MT concentration, even if you follow a previously reported protocol: this would allow readers to have this information right away.
Surgery for cannula implantation
A better explanation of the procedure employed should be presented, as reported, for example, by Souza, et al. Indoleamine-2,3-dioxygenase mediates neurobehavioral alterations induced by an intracerebroventricular injection of amyloid-β1-42 peptide in mice. Brain, Behavior, and Immunity 2016; 56: 363-377-
In this section the microinjection in the prelimbic and intra limbic areas reported in Figure 2 B should be described.
Nissl staining
It is not clear when this experiment was conducted and how many animals underwent this procedure. Please report.
Experimental design
This section is confusing. A better explanation of the two sequential experiments should be provided: e.g, experiment 1 was carried out to monitor behavioral changes after STZ injection through a battery of behavioral tests, etc… Experiment 2 was performed to evaluate molecular and cellular changes in the brain at the time of the onset of depression-like signs. This is important also for ethical issues.
Page 18 line 540: please write: “ After the surgical intervention, animal body weight…”.
Estimation of kynurenine metabolites
Please change into “Estimation of serotonin and kynurenine shunt metabolites”
Which chromatographic system was employed? Did the authors used an HPLC or an U(H)PLC chromatograph? Chromatographic time seems quite short for HPLC: Why didn't you evaluate quinolinic acid, which is a excitogenic kynurenine metabolite? This should be mentioned in the manuscript as a limitation of this study.
Enzyme-linked immunosorbent assay (ELISA)
Please mention the total protein assay method used to report results as pg/mg protein.
Data and statistical analysis
Please indicate the comparison test used, when independent and when pairwise, if performed (this is not clear).
Results
Please carefully check the figures and captions so that the caption content can be read independently of the text. Please check all statistics.
Discussion
In this section the authors should provide a deeper description of 1-MT, its applications and past investigations as an IDO inhibitor, reporting and citing the main studies conducted with this compound in order to promote Kyn shunt manipulations together the related clinical applications.
Please cite, when presenting the Kyn shunt and its involvement in psychiatric disorders, the studies of: Muneer A. Kynurenine Pathway of Tryptophan Metabolism in Neuropsychiatric Disorders: Pathophysiologic and Therapeutic Considerations. Clin Psychopharmacol Neurosci. 2020; 18(4):507-526; Carpita B. et al. Kynurenine pathway and autism spectrum phenotypes: an investigation among adults with autism spectrum disorder and their first-degree relatives. CNS Spectr. 2023;28(3):374-385.
The discussion section should try to provide some hypotheses about the area-dependent differences found, despite common effect induced by 1-MT, while also reporting the importance to carry out further investigation on this topic at the therapeutic levels. Moreover, it is important that the authors highlight that the work could be useful in unraveling the molecular and cellular bases of depression symptoms in Alzheimer's Disease, not only focused on the understanding of new mechanisms for treating depression "per se". Discussion and conclusions should be revised to broaden the scientific impact of the study and to justify this procedure that, even if approved by the Ethical Committee, sounds invasive to animals.
At the end of the Discussion section, limitations of the study should be also indicated.
Comments on the Quality of English LanguageThere are some inaccuracies in English language to check.
Author Response
Dear Reviewer:
Thank you for your comments concerning our manuscript entitled “Inhibition of indoleamine 2,3-dioxygenase in the prelimbic and infralimbic cortices exerts antidepressant effects through different pathways in ICV-STZ rats” (ID: ijms- 3010257). Those comments are all valuable and very helpful for revising and improving our paper, as well as the important guiding significance to us researches. We have studied comments carefully and have made correction which we hope meet with approval. In addition, our manuscript has undergone English language editing by English Editing Department of MDPI. Revised portion are marked in highlight in the manuscript. The main corrections in the paper and the responds to the reviewer's comments are as follow.
Comments 1: Title: I suggest to improve the title as follows: “Inhibition of indoleamine 2,3-dioxygenase in the prelimbic and infralimbic cortices exerts antidepressant-like effects through different pathways in rats under intracerebroventricular injection with streptozotocin”
Response 1: Thank you for pointing this out. We revised the title as your suggestion.
Comments 2: Abbreviations: Due to the high number of abbreviations used in the manuscript and for a better understanding of the text, I suggest the authors to create an Abbreviations sections at the beginning or at the end of the manuscript, on the bases of the IJMS guidelines.
Response 2: Thank you for pointing this out. We added an Abbreviations section at the end of the manuscript.
Comments 3: Abstract: Abbreviations “Iba-1 positive” and “1-MT” should be first presented as full names with abbreviations in parenthesis.
Response 3: Thank you for pointing this out. We have added definitions for GFAP, Iba-1, and 1-MT which are highlighted in the revised version.
Comments 4&5: Introduction: Alzheimer’s disease should be better presented; Introduction: Depression symptoms in Alzheimer’s disease should be better treated, possibly considering their features in respect to non-neurological depression (see, among others, these articles: Aguera-Ortiz et al. Depression in Alzheimer's Disease: A Delphi Consensus on Etiology, Risk Factors, and Clinical Management. Front. Psychiatry, Sec. Aging 2021; 12:1-16; Modrego PJ, de Cerio LD, Lobo A. The Interface between Depression and Alzheimer's Disease. A Comprehensive Approach. Ann Indian Acad Neurol. 2023;26(4):315-325).
Response 4 and 5: Thank you for pointing this out. After careful consideration of the comments 4 and 5 and recommendations, we have revised the first paragraph and cited these references.
Alzheimer's disease (AD), a common neurodegenerative disorder, is the most prevalent form of senile dementia, causing progressive deterioration of cognition, behavior, and rational skills. Neuropsychiatric symptoms like depression often accompany and/or precedes AD onset [1, 2]. Depression is regarded as an early symptom of AD or a risk factor for AD with a negative impact on the quality of life of patients and caregivers. However, the pathophysiological mechanisms underlying depression in AD patients remain less well defined.
Comments 6: Introduction: Page 1 line 38: sporadic AD (sAD) – present this AD form through a short sentence;
Response 6: Thank you for pointing this out. We have revised this sentence as follow:
Over 95% of AD cases manifest as sporadic AD (sAD), influenced by the intricate interplay of genetic and environmental factors.
Comments 7: Introduction: Page 1 line 40: Remove “Recently”;
Response 7: Thank you for pointing this out. We have removed the “Recently” in line 40.
Comments 8: Introduction: Page 2 lines 47-49: please change as: “Furthermore, ICV-STZ also reproduces the AD progression, promoting first non-cognitive features such as depression-like behaviors in rodents”
Response 8: Thanks for your suggestion. We've replaced this sentence.
Comments 9: Page 2 lines 68-72: As aforementioned, this part should be removed from the Introduction section, since no results should be presented here. The authors should insert it in the Result and Discussion sections.
Response 9: Thank you. Indeed, here we hope that a brief description of the results leads to the following introduction of the PrL and IL.
Comments 10: Animals: Please indicate the animals’ age. Describe food type and animal nutritional conditions. It is also important to report the total number of rats included into the study, which, due to the invasive procedure employed, must be as small as possible. The authors should also report all efforts and procedures carried out to contain any possible animal suffering.
Response 10: Thank you for pointing this out. We have revised “4.1 Animals” paragraph as follow:
A total of 184 adult male Sprague‒Dawley rats (8 ± 1 weeks old, 250-280 g) were procured from the Animal Center of Peking University (Beijing, China). All rats were housed individually in plastic cages with ad libitum access to food and water at an optimum temperature (25 ± 2) °C and 55%-65% relative humidity. We chose the regular maintenance diet for rats, which ensured that they were well nourished. The bottle was positioned in the middle of the cage cover to eliminate location bias. A 12 h/12 h light/dark cycle (lights on at 9:00 AM) was regulated in the animal house. All of the rats were allowed to acclimate for 7 d before receiving any experimental manipulation. All of the experimental procedures complied with the guidelines of the "Animal Research: Reporting of In Vivo Experiments (ARRIVE)" [65] and carried out in accordance with the National Re-search Council's Guide for the Care and Use of Laboratory Animals. The animal experiment protocol was approved by the Peking University Committee on Animal Care and Use (permission no. LA 2020279). During the experiments, rats may suffer pain from surgery, forced swimming test, and other experiments. Therefore, we try to minimize the pain and stress that the rats may suffer, and the care process is described in detail in each of the next experimental method.
Comments 11: Drugs and treatments: page 17 line 500: what was the starting concentration of 1-methyl-tryptophan? Was 1-MT a racemic mixture or the L-isomer? Please report these points. Please report also the final 1-MT concentration, even if you follow a previously reported protocol: this would allow readers to have this information right away
Response 11: Thank you for pointing this out. We have revised this paragraph as follow:
The racemic mixture of 1-MT (1-Methyl-DL-tryptophan; Sigma‒Aldrich, Cat# 860646, MO, USA) was dissolved in 1 M HCl and adjusted the pH to 6.5 using NaOH, and a stock solution at a concentration of 1 mg/mL was prepared with 0.9% sterile sodium chloride solution [69]. Then, it was diluted using 0.9% sterile sodium chloride solution to the final treatment concentration of 50 μg/mL, according to antidepressant effective dose of 1-MT [70]. 1-MT was injected once daily into the PrL or IL bilaterally (0.2 μL per side) at a rate of 0.1 μL/min at 9:00-11:00 AM from d 1 to d 8. The injection cannula was kept in place for another 2 min to allow the drug to diffuse from the tip entirely. The vehicle rats were administered with an equal volume of 0.9% sterile sodium chloride solution.
Comments 12: Surgery for cannula implantation: A better explanation of the procedure employed should be presented, as reported, for example, by Souza, et al. Indoleamine-2,3-dioxygenase mediates neurobehavioral alterations induced by an intracerebroventricular injection of amyloid-β1-42 peptide in mice. Brain, Behavior, and Immunity 2016; 56: 363-377-In this section the microinjection in the prelimbic and intra limbic areas reported in Figure 2 B should be described.
Response 12: Thank you for pointing this out. We added the following sentence at the end of the method 4.2:
The injection placements were verified by unaided eye or Nissle staining and showed in Figure 2B and Figure 8A.
Comments 13: Nissl staining: It is not clear when this experiment was conducted and how many animals underwent this procedure. Please report.
Response 13: Thank you for pointing this out. For perfusion-treated rats, we stained the cannula sections of each rat with Nissl staining. However, some of the rat brain tissues were used in Western-blot, ELISA and HPLC-ECD. In these biochemical testing, brain tissue should be fresh and couldn’t be fixed with paraformaldehyde. Technically, the needle placement in these rats couldn’t be detect by histological tissue section. Therefore, we observed the needle injection site and canular placement with the unaided eye when we were collecting brain tissue. We put the whole brain on the brain matrix. Then, we cut down brain coronal sections from bregma -2 ~ 0 mm to observe the placements of the cannulae in the lateral ventricles, and we cut coronal sections from bregma 2 ~ 4 mm to observe the placements of the cannulae in the PrL and IL.
Comments 14: Experimental design: This section is confusing. A better explanation of the two sequential experiments should be provided: e.g, experiment 1 was carried out to monitor behavioral changes after STZ injection through a battery of behavioral tests, etc… Experiment 2 was performed to evaluate molecular and cellular changes in the brain at the time of the onset of depression-like signs. This is important also for ethical issues.
Response 14: Thank you for pointing this out. We have added explanations at the beginning of this section as your suggestions.
Comments 15: Experimental design: Page 18 line 540: please write: “After the surgical intervention, animal body weight…”.
Response 15: Thank you for pointing this out. We have revised this problem.
Comments 16: Estimation of kynurenine metabolites: Please change into “Estimation of serotonin and kynurenine shunt metabolites”.
Response 16: Thank you for pointing this out. We have changed the title to “Estimation of serotonin and kynurenine metabolites”.
Comments 17: Estimation of kynurenine metabolites: Which chromatographic system was employed? Did the authors used an HPLC or an U(H)PLC chromatograph? Chromatographic time seems quite short for HPLC: Why didn't you evaluate quinolinic acid, which is a excitogenic kynurenine metabolite? This should be mentioned in the manuscript as a limitation of this study.
Response 17: Thank you for pointing this out. An HPLC or an UHPLC method mainly depends on the used chromatographic column. Here, after optimization, we chose an Ultimate XB-C18 column with 5 μm packing material size. Traditionally, a method using chromatographic column with 5 μm packing material size and system pressure lower than 6000 psi belongs to a HPLC method. To be rigorous, we modified corresponding statement in the manuscript. In addition, we have tried to detect the level of quinolinic acid, but the peak of quinolinic acid always has a very serious trailing phenomenon, so the detection method of quinolinic acid needs to be further optimized in the future. We added this at the end of the fourth paragraph of the discussion.
Comments 18: Enzyme-linked immunosorbent assay (ELISA): Please mention the total protein assay method used to report results as pg/mg protein.
Response 18: Thank you for pointing this out. We have revised this paragraph as follow:
Rat ELISA kits were used to measure the levels of IL-1β (ABclonal, Cat# RK00009, Wuhan, China), IL-6 (ABclonal, Cat# RK00020, Wuhan, China), and TNF-α (ABclonal, Cat# RK00029, Wuhan, China) in the PrL and IL. Briefly, the tissue was homogenized and centrifuged at 2000 ×g for 20 min at 4 °C; then, the supernatant was extracted and the total protein in the samples was quantified by BCA kit (Thermo, Cat# 23227, MA, USA). Another 100 µL of supernatant was incubated in a 96-well plate. The cytokine levels were estimated by interpolation from a standard curve by colorimetric measurements at 450 nm (correction wavelength 630 nm) on an ELISA plate reader (Thermo Fisher Scientific Mul-tiskan Mk3, MA, USA). Each data point comes from an individual rat. The results are shown as pg/mg of protein.
Comments 19: Data and statistical analysis: Please indicate the comparison test used, when independent and when pairwise, if performed (this is not clear).
Response 19: Thank you for pointing this out. As your suggestion, we have added the following sentence in this paragraph: The types of comparison test involved in each experiment was labeled in the figure legends.
Comments 20: Results: Please carefully check the figures and captions so that the caption content can be read independently of the text. Please check all statistics.
Response 20: Thank you for pointing this out. We have examined all the figures, captions, and statistics in the manuscript.
Comments 21: Discussion: In this section the authors should provide a deeper description of 1-MT, its applications and past investigations as an IDO inhibitor, reporting and citing the main studies conducted with this compound in order to promote Kyn shunt manipulations together the related clinical applications. Please cite, when presenting the Kyn shunt and its involvement in psychiatric disorders, the studies of: Muneer A. Kynurenine Pathway of Tryptophan Metabolism in Neuropsychiatric Disorders: Pathophysiologic and Therapeutic Considerations. Clin Psychopharmacol Neurosci. 2020; 18(4):507-526; Carpita B. et al. Kynurenine pathway and autism spectrum phenotypes: an investigation among adults with autism spectrum disorder and their first-degree relatives. CNS Spectr. 2023;28(3):374-385.
The discussion section should try to provide some hypotheses about the area dependent differences found, despite common effect induced by 1-MT, while also reporting the importance to carry out further investigation on this topic at the therapeutic levels. Moreover, it is important that the authors highlight that the work could be useful in unraveling the molecular and cellular bases of depression symptoms in Alzheimer's Disease, not only focused on the understanding of new mechanisms for treating depression "per se". Discussion and conclusions should be revised to broaden the scientific impact of the study and to justify this procedure that, even if approved by the Ethical Committee, sounds invasive to animals.
At the end of the Discussion section, limitations of the study should be also indicated.
Response 21: Thank you for pointing this out. We have cited your suggested references and revised the discussion section. We added the following sentences in the second paragraph of the discussion:
1-MT has been confirmed to have antidepressant effects via multiple routes of admin-istration in several animal models [43, 44]. Subcutaneous administration of the IDO in-hibitor 1-MT was shown to improve depression-like behaviors in ICV-STZ mice [18]. However, the mechanism of action of 1-MT in the central nervous system, especially in subregions of the mPFC, has not been studied.
We added the following sentences at the end of the discussion:
Although 1-MT injections in both PrL and IL had antidepressant effects, we observed region-dependent differences in these subregions, suggesting the importance of further investigation on this topic at the therapeutic level.
The limitation about quinolinic acid have been added at the end of fourth paragraph of the discussion.
Furthermore, we have tried to detect the level of quinolinic acid, but the peak of quinolinic acid always has a very serious trailing phenomenon, so the detection method of quinolinic acid needs to be further optimized in the future.

Reviewer 2 Report (New Reviewer)
Comments and Suggestions for Authors
The manuscript is generally well designed but the authors need to revise english probably a native speaker can do the job.
lots of abbreviations are not defined
please avoid using "and" in the beginning of sentence
Please use past tense to describe the results
Could the authors justify their usage of STZ as a model?
Have the authors measured brain glucose conc or any of markers related to disturbances in glucose metabolism ?
Have the authors detected cortisol level? probably can explain the symptoms related to depression
Comments on the Quality of English Language
moderate english editing is required
Author Response
Dear Reviewer:
Thank you for your comments concerning our manuscript entitled “Inhibition of indoleamine 2,3-dioxygenase in the prelimbic and infralimbic cortices exerts antidepressant effects through different pathways in ICV-STZ rats” (ID: ijms- 3010257). Those comments are all valuable and very helpful for revising and improving our paper, as well as the important guiding significance to us researches. We have studied comments carefully and have made correction which we hope meet with approval. In addition, our manuscript has undergone English language editing by English Editing Department of MDPI. Revised portion are marked in highlight in the manuscript. The main corrections in the paper and the responds to the reviewer's comments are as follow.
Comments 1: lots of abbreviations are not defined.
Response 1: Thank you for pointing this out. We have added definitions for GFAP and Iba, which are highlighted in the revised version.
Comments 2: please avoid using "and" in the beginning of sentence.
Response 2: Thanks for your suggestion. As suggested by the reviewer, we have revised the problem like this, which are highlighted in the revised version.
Comments 3: Please use past tense to describe the results
Response 3: Thanks for your suggestion. We have revised the tenses of the manuscript.
Comments 4: Could the authors justify their usage of STZ as a model?
Response 4: At the beginning of the 21st century, scientists proposed the concept of AD as a type 3 (brain-specific) diabetes based on the clinical observations of AD patients that glucose hypometabolism is an early and persistent sign of AD and that Alzheimer’s brains present features of impaired insulin signaling. Therefore, ICV-STZ injections are exploited by some investigators as a non-transgenic model of this disease and used for preclinical testing of pharmacological therapies for AD [PMID: 25744568]. Other AD phenomena associated with the ICV-STZ model include oxidative stress, mitochondrial dysfunction, cholinergic dysfunction, and neuroinflammation [PMID: 36591205]. To date, the ICV-STZ model has been used in a large number of articles in studies of AD.
On the other hand, there is a very high failure rate in clinical trials of AD therapeutic drugs (approximately 99.6%), many of which have been successful in preclinical trials using transgenic animal models, thus leading to questions about the validity of existing transgenic models [PMID: 28025715].
In summary, we chose the ICV-STZ model in this study.
Comments 5: Have the authors measured brain glucose conc or any of markers related to disturbances in glucose metabolism?
Response 5: We did not measure brain glucose conc or any of markers related to disturbances in glucose metabolism in this study. Previous studies have demonstrated disturbances in glucose metabolism in the brain of ICV-STZ rats, as evidenced by impaired insulin signaling, overactivation of glycogen synthase kinase-3β, decreased levels of major brain glucose transporters, and downregulated protein O-GlcNAcylation [PMID: 25088942 and 24380887]. These disturbances in glucose metabolism can cause Aβ deposition and Tau hyperphosphorylation, which finally cause learning and cognitive impairments [PMID: 25744568]. In our previous studies [PMID: 29080930 and 33352241], we have found that ICV-STZ rats can develop cognitive deficits on both d14 and d21, and increased Aβ expression and elevated Tau phosphorylation were also observed in the hippocampus.
Comments 6: Have the authors detected cortisol level? probably can explain the symptoms related to depression
Response 6: We did not examine corticosterone and cortisol levels in ICV-STZ rats in this study. It has been shown that plasma corticosterone levels were significantly increased in STZ-induced diabetic rodents [PMID: 16722465], which may be one of the pathological mechanisms of diabetes-related depression. However, in ICV-STZ rats, researchers did not observe changes in plasma corticosterone levels on day 21 [PMID: 35027835]. Therefore, whether the mechanism of depression-like behaviors in ICV-STZ rats is related to the activation of the HPA axis need further studies.
Response to Comments on the Quality of English Language
Thank you for pointing this out. We have purchased the MDPI’s rpaid English editing service recommended by IJMS, and this manuscript have revised by an experienced English-speaking editor. Revisions are highlighted in the new submission.

Reviewer 3 Report (Previous Reviewer 1)
Comments and Suggestions for Authors
1) In the resubmitted manuscript the changes introduced by the authors are not marked. Please, mark the changes clearly, e.g. using colour marking.
2) The language of manuscript still requires intensive editing. The whole manuscript must be corrected by a native speaker.
Below only some example of odd sentences.
Abstract, line 10 "An intracerebroventricular injection of streptozotocin (ICV-STZ) is used to simulate sporadic Alzheimer’s disease (sAD) in rats." - rats do NOT develop Alzheimer's disease - they are used to model the disease
line 11 "Rats exhibit depression-like behaviors at the beginning of this model." - where is the beginning of the model?
line 18 as evidenced by a decrease in kynurenic acid and kynurenine aminotransferase II accompanied by defects in astrocytes, reflected by decreases in GFAP-positive cells and glial transporters and morphological damage ??? - enzymes do not increase, defect is not an appropriate word, etc.
line 21 an increase in 3-hydroxy-kynurenine and kynurenine 3-monooxygenase - again - in case of metabolite, the level increases, whereas in the case of enzyme - the activity
Comments on the Quality of English LanguageThe language of manuscript still requires intensive editing. The whole manuscript must be corrected by a native speaker.
Below only some example of odd sentences.
Abstract, line 10 "An intracerebroventricular injection of streptozotocin (ICV-STZ) is used to simulate sporadic Alzheimer’s disease (sAD) in rats." - rats do NOT develop Alzheimer's disease - they are used to model the disease
line 11 "Rats exhibit depression-like behaviors at the beginning of this model." - where is the beginning of the model?
line 18 as evidenced by a decrease in kynurenic acid and kynurenine aminotransferase II accompanied by defects in astrocytes, reflected by decreases in GFAP-positive cells and glial transporters and morphological damage ??? - enzymes do not increase, defect is not an appropriate word, etc.
line 21 - an increase in 3-hydroxy-kynurenine and kynurenine 3-monooxygenase - again - in case of metabolite, the level increases, whereas in the case of enzyme - the activity
Author Response
Dear Reviewer:
Thank you for your comments concerning our manuscript entitled “Inhibition of indoleamine 2,3-dioxygenase in the prelimbic and infralimbic cortices exerts antidepressant effects through different pathways in ICV-STZ rats” (ID: ijms- 3010257). Those comments are all valuable and very helpful for revising and improving our paper, as well as the important guiding significance to us researches. We have studied comments carefully and have made correction which we hope meet with approval. In addition, our manuscript has undergone English language editing by English Editing Department of MDPI. Revised portion are marked in highlight in the manuscript. The main corrections in the paper and the responds to the reviewer's comments are as follow.
Point-by-point response to Comments and Suggestions for Authors
Comments 1: In the resubmitted manuscript the changes introduced by the authors are not marked. Please, mark the changes clearly, e.g. using colour marking.
Response 1: Thank you for pointing this out. In the resubmitted manuscript, we have labeled the changes using highlighting.
Comments 2: The language of manuscript still requires intensive editing. The whole manuscript must be corrected by a native speaker.
Response 2: Thank you for pointing this out. We have purchased the MDPI’s rpaid English editing service recommended by IJMS, and this manuscript have revised by an experienced English-speaking editor. Revisions are highlighted in the new submission.
Response to Comments on the Quality of English Language
Point 1: Abstract, line 10 "An intracerebroventricular injection of streptozotocin (ICV-STZ) is used to simulate sporadic Alzheimer’s disease (sAD) in rats." -rats do NOT develop Alzheimer's disease - they are used to model the disease
Response 1: Thanks for your suggestion. We have revised this sentence as follows: An intracerebroventricular injection of streptozotocin (ICV-STZ) is used as an animal model of sporadic Alzheimer’s disease (sAD).
Point 2: line 11 "Rats exhibit depression-like behaviors at the beginning of this model." - where is the beginning of the model?
Response 2: Thanks for your suggestion. We have revised this sentence as follows: Rats exhibit depression-like behaviors at the beginning (day 7) of this model.
Point 3: line 18 as evidenced by a decrease in kynurenic acid and kynurenine aminotransferase II accompanied by defects in astrocytes, reflected by decreases in GFAP-positive cells and glial transporters and morphological damage ??? - enzymes do not increase, defect is not an appropriate word, etc.
Response 3: Thanks for your suggestion. We have revised this sentence as follow: as evidenced by a decrease in kynurenic acid level and kynurenine aminotransferase II expression accompanied byabnormalities in astrocytes, reflected by decreases in GFAP-positive cells and glial transporters and morphological damage
Point 4: line 21 - an increase in 3-hydroxy-kynurenine and kynurenine 3-monooxygenase - again - in case of metabolite, the level increases, whereas in the case of enzyme - the activity
Response 4: Thanks for your suggestion. We have revised this sentence as follow: an increase in 3-hydroxy-kynurenine level and kynurenine 3-monooxygenase expression

Round 2
Reviewer 1 Report (New Reviewer)
Comments and Suggestions for Authors
Thank you for your point-by-point reply and for providing an improved version of the manuscript, accordingly to my suggestions. Anyway, this revised version of the manuscript ijms-3010257 still requires extra revision at this stage of the peer review process. I apologize for that, but some aspects and parts of the manuscript remain still to be better described or explained. My additional criticisms and suggestions raise with the aim to gradually improve this work to the best possible.
General comments:
The manuscript contains many results, is quite complicated to read and, for the same reason, to be reviewed. I overall recommend revising the manuscript trying to clarify it by keeping the description of methods and results as straighten and informative as possible. In addition to this general advice, authors will find here further specific suggestions for improving the drafting of the Abstract, Introduction and Discussion sections. Some critical points are also indicated in the Methods and Results sections.
The authors have changed the title as I previously suggested: the new title provided is quite correct, but English language still needs to be revised. By revising English and after a careful reflection, I rather propose the following one, or similar, for a proper introduction to the specific topic of the paper:
“Inhibition of indoleamine 2,3-dioxygenase exerts anti-depressant-like effects through distinct pathways in pre-limbic and infralimbic cortices after intracerebroventricular streptozotocin injections reproducing sporadic Alzheimer’s Disease in rats.”
Specific comments:
Abstract
This section is very important to present and resume the whole investigation.
The authors should change following these suggestions:
Page 1, lines 11-16: The application of intracerebroventricular injections of streptozotocin (ICV-STZ) is considered an useful animal model to mimic the onset and progression of sporadic Alzheimer’s Disease.
In rodents, animals exhibit depression-like behaviors after 7 days from this procedure. Indoleamine 2,3 dioxygenase (IDO), a rate-limiting…., has been closely related to depression and AD.
Present study aimed to investigate the pathophysiological mechanisms of preliminary depression-like behaviors in ICV-STZ rats in two distinct cerebral regions, the prelimbic cortex (PrL) and the infralimbic cortex (IL), both presumably involved in AD progression in this model, with a focus on..
Page 1, lines 16-24: Results showed an increased Kyn/Trp ratio in both the PrL and IL of ICV-STZ rats, but, intriguingly, abnormalities in downstream metabolic pathways were different, being associated with distinct biological effects. In the PrL, the neuroprotective branch of the kynurenine (Kyn) pathway was attenuated, as evidenced by a decrease in kynurenic acid level and Kyn aminotransferase II expression, accompanied by astrocyte alterations, such as the decrease of glial fibrillary acidic protein (GFAP)-positive cells and increase of morphological damage. In the IL, the excitogenic branch of the Kyn pathway was enhanced, as evidenced by an increase in 3-hydroxy-kynurenine (3-HK) level and kynurenine 3-monooxygenase (KMO) expression…
Page 1, lines 28-31: These results suggest that the antidepressant-like effects linked to Trp metabolism changes induced by 1-MT in the PrL and IL, occur through different pathways, precisely by enhancing anti-excitatory effects in the PrL and attenuating excitogenic responses in the IL, involving distinct glial cells.
Introduction
The Introduction section has been improved, nevertheless at page 2 lines 74-78, there is still the sentence that should be moved in the Results or in the Discussion section. I think that placing this sentence later in the text is quite less confusing for readers. Moreover, in this section you should provide only those previous results or evidences that prompted you to investigate these brain regions specifically.
Please, avoid the use of expressions like “obviously”.
Discussion
More explanation is still required for the lack of changes at the level of IDO expression and the lack of changes of serotonin levels in the investigated regions. Indeed, these are important points: present results might reflect the physiological and pathogenetic roles of Trp metabolism, different Kyn shunt branches and their compartmentalization or relationships in different brain regions. Mechanisms presumably involved in these negative results should be indicated.
Please, also discuss the results always in terms of depression-like or AD-related depression, since in depression there is not inevitably an evolution into AD, as occurs instead in the investigated model.
The lack of determination of quinolinic acid still remain a main limitation of the study. The sentence at page 16, lines 444-447, should be changed and moved at the end of the Discussion. For instance, I suggest: “A main issue of the present study was the absence of the measure of quinolinic acid levels, due to methodological concerns. Consequently, one of the future developments of the present investigation will be the analysis of this important Kyn metabolite, in order to clarify its possible involvement in the observed changes in the ICV-STZ rat model, and its potential role in depression signs linked to AD evolution”.
Also, there is still the need to better highlight the importance of the obtained results for understanding and unraveling the mechanisms underlying depression-like signs linked to AD.
The manuscript’s sections Methods and Results still require careful revision.
Results
Page 3 line 108: change into “Additionally”.
Figures containing histograms should be clearer: use lines to indicate which are the significant comparisons obtained, to make it clear which groups were compared when the statistical significance was attained. It is still quite confusing as it is now.
There is really a huge quantity of data that are certainly not easy to present, but quantitative results found for the parameters evaluated in the investigation must be presented and they should be clearly included (means ± SEMs, unit of measurement) in the manuscript, or in the text or in the figure legends. An alternative way can be also to separately display tables with the means ± SEMs (unit of measurement) as Supplementary material.
Supplementary data: if results are not significant, it is not necessary to report the statistical significance applied. Just report the statistical method used. This information should be rather included in the Method section - Data and Statistical Analysis.
Methods
First of all: the number of animals employed is elevated. The experimental design does not unambiguously explain the issue. The authors must attest this number or provide more explanation about that, in relation to the statistics applied and the targets of the study.
4.5. Experimental design: I suggest to move this section after the section 4.1. Animals . This is important to immediately inform readers, for the sake of clarity.
The chromatographic procedure used to separate and measure Trp, 5-HT and Kyn metabolites still needs further clarification. First: has the method described in the paper been validated before? Is there a literature reference? If this is the case, the authors must cite it, eventually indicating modifications employed. Otherwise, the chromatographic conditions presented appear still incomplete, so that the authors should provide more details about that: for instance, the working pressure values; they should also explain the method’s resolution and the theoretical plate specifics in respect to the reported low flow rate and HPLC conditions. The calibration line concentrations and IS retention times are missing. The Supplementary material should include some typical chromatograms.
These same questions (has the method described in the paper been validated before? Is there a literature reference?) are valuable for all the other techniques and methodologies used in this work, except for the commercially purchased ones. Please verify.
Comments on the Quality of English Language
English language must be reviewed by a native speaker or expert
Author Response
General comments:
The manuscript contains many results, is quite complicated to read and, for the same reason, to be reviewed. I overall recommend revising the manuscript trying to clarify it by keeping the description of methods and results as straighten and informative as possible. In addition to this general advice, authors will find here further specific suggestions for improving the drafting of the Abstract, Introduction and Discussion sections. Some critical points are also indicated in the Methods and Results sections.
The authors have changed the title as I previously suggested: the new title provided is quite correct, but English language still needs to be revised. By revising English and after a careful reflection, I rather propose the following one, or similar, for a proper introduction to the specific topic of the paper:
“Inhibition of indoleamine 2,3-dioxygenase exerts anti-depressant-like effects through distinct pathways in prelimbic and infralimbic cortices after intracerebroventricular streptozotocin injections reproducing sporadic Alzheimer’s Disease in rats.”
Response: Thanks for your suggestion. We changed the title as follow:
Inhibition of indoleamine 2,3-dioxygenase exerts antidepressant-like effects through distinct pathways in prelimbic and infralimbic cortices in rats under intracerebroventricular injection with streptozotocin.
Specific comments:
Abstract
This section is very important to present and resume the whole investigation.
The authors should change following these suggestions:
Page 1, lines 11-16: The application of intracerebroventricular injections of streptozotocin (ICV-STZ) is considered an useful animal model to mimic the onset and progression of sporadic Alzheimer’s Disease.
In rodents, animals exhibit depression-like behaviors after 7 days from this procedure. Indoleamine 2,3 dioxygenase (IDO), a rate-limiting…., has been closely related to depression and AD.
Present study aimed to investigate the pathophysiological mechanisms of preliminary depression-like behaviors in ICV-STZ rats in two distinct cerebral regions, the prelimbic cortex (PrL) and the infralimbic cortex (IL), both presumably involved in AD progression in this model, with a focus on..
Page 1, lines 16-24: Results showed an increased Kyn/Trp ratio in both the PrL and IL of ICV-STZ rats, but, intriguingly, abnormalities in downstream metabolic pathways were different, being associated with distinct biological effects. In the PrL, the neuroprotective branch of the kynurenine (Kyn) pathway was attenuated, as evidenced by a decrease in kynurenic acid level and Kyn aminotransferase II expression, accompanied by astrocyte alterations, such as the decrease of glial fibrillary acidic protein (GFAP)-positive cells and increase of morphological damage. In the IL, the excitogenic branch of the Kyn pathway was enhanced, as evidenced by an increase in 3-hydroxy-kynurenine (3-HK) level and kynurenine 3-monooxygenase (KMO) expression…
Page 1, lines 28-31: These results suggest that the antidepressant-like effects linked to Trp metabolism changes induced by 1-MT in the PrL and IL, occur through different pathways, precisely by enhancing anti-excitatory effects in the PrL and attenuating excitogenic responses in the IL, involving distinct glial cells.
Response: Thanks for your suggestions. We revised Abstract according to your suggestions:
The application of intracerebroventricular injection of streptozotocin (ICV-STZ) is considered as an useful animal model to mimic the onset and progression of sporadic Alzheimer’s disease (sAD). In rodents, at 7 days of the experiment, the animals exhibit depression-like behaviors. Indoleamine 2,3-dioxygenase (IDO), a rate-limiting enzyme catalyzing the conversion of tryptophan (Trp) to kynurenine (Kyn), is closely related to depression and AD. Present study aimed to investigate the pathophysiological mechanisms of preliminary depression-like behaviors in ICV-STZ rats in two distinct cerebral regions of the medial prefrontal cortex, the prelimbic cortex (PrL) and infralimbic cortex (IL), both presumably involved in AD progression in this model, with a focus on IDO-related Kyn pathways. Results showed an increased Kyn/Trp ratio in both the PrL and IL of ICV-STZ rats, but, intriguingly, abnormalities in downstream metabolic pathways were different, being associated with distinct biological effects. In the PrL, the neuroprotective branch of the Kyn pathway was attenuated, as evidenced by a decrease in kynurenic acid (KA) level and Kyn aminotransferase II (KAT II) expression, accompanied by astrocyte alterations, such as the decrease of glial fibrillary acidic protein (GFAP)-positive cells and increase of morphological damage. In the IL, the neurotoxic branch of the Kyn pathway was enhanced, as evidenced by an increase in 3-hydroxy-kynurenine (3-HK) level and kynurenine 3-monooxygenase (KMO) ex-pression paralleled by the overactivation of microglia, reflected by an increase in ionized cal-cium binding adaptor molecule 1 (Iba1)-positive cells and cytokines with morphological altera-tions. Synaptic plasticity was attenuated in both subregions. Additionally, microinjection of the selective IDO inhibitor 1-Methyl-DL-tryptophan (1-MT) in the PrL or IL alleviated depression-like behaviors by reversing these different abnormalities in the PrL and IL. These results suggest that the antidepressant-like effects linked to Trp metabolism changes induced by 1-MT in the PrL and IL, occur through different pathways, specifically by enhancing neuroprotective branch in the PrL and attenuating neurotoxic branch in the IL, involving distinct glial cells.
Introduction
The Introduction section has been improved, nevertheless at page 2 lines 74-78, there is still the sentence that should be moved in the Results or in the Discussion section. I think that placing this sentence later in the text is quite less confusing for readers. Moreover, in this section you should provide only those previous results or evidences that prompted you to investigate these brain regions specifically.
Please, avoid the use of expressions like “obviously”.
Response: Thanks for your suggestion. We deleted these sentences.
Discussion
More explanation is still required for the lack of changes at the level of IDO expression and the lack of changes of serotonin levels in the investigated regions. Indeed, these are important points: present results might reflect the physiological and pathogenetic roles of Trp metabolism, different Kyn shunt branches and their compartmentalization or relationships in different brain regions. Mechanisms presumably involved in these negative results should be indicated.
Please, also discuss the results always in terms of depression-like or AD-related depression, since in depression there is not inevitably an evolution into AD, as occurs instead in the investigated model.
The lack of determination of quinolinic acid still remain a main limitation of the study. The sentence at page 16, lines 444-447, should be changed and moved at the end of the Discussion. For instance, I suggest: “A main issue of the present study was the absence of the measure of quinolinic acid levels, due to methodological concerns. Consequently, one of the future developments of the present investigation will be the analysis of this important Kyn metabolite, in order to clarify its possible involvement in the observed changes in the ICV-STZ rat model, and its potential role in depression signs linked to AD evolution”.
Also, there is still the need to better highlight the importance of the obtained results for understanding and unraveling the mechanisms underlying depression-like signs linked to AD.
Response: Thank you for pointing this out. We added the following sentences at the end of the second paragraph of the discussion:
However, neither ICV-STZ nor 1-MT treatment affected IDO expression in the PrL and IL. These results are consistent with previous studies in diabetic and LPS mice, in which 1-MT significantly alleviated depression-like behaviors but did not alter IDO expression [45, 46]. In addition, although classical animal models of depression have shown reduced 5-HT levels in the PFC [47], this has not been observed in ICV-STZ rats. These finding suggest that AD-related depression may have a unique pathogenesis.
In addition, we have moved the limitations at the end of the Discussion and highlighted them.
The manuscript’s sections Methods and Results still require careful revision.
Results
Page 3 line 108: change into “Additionally”.
Response: Thank you for pointing this out. We have revised this word.
Figures containing histograms should be clearer: use lines to indicate which are the significant comparisons obtained, to make it clear which groups were compared when the statistical significance was attained. It is still quite confusing as it is now.
Response: Thank you for pointing this out. We have revised our figure legends to better clarify the meaning of significant symbols:
“*p < 0.05, **p < 0.01 vs. vehicle group” was revised to “*p < 0.05, **p < 0.01 vs. Vehicle + Vehicle group”.
“#p < 0.05, ##p < 0.01 vs. ICV-STZ group” was revised to “*p < 0.05, **p < 0.01 vs. ICV-STZ + Vehicle group.”
Due to limited space of figures, lines were not used as significant indicators.
There is really a huge quantity of data that are certainly not easy to present, but quantitative results found for the parameters evaluated in the investigation must be presented and they should be clearly included (means ± SEMs, unit of measurement) in the manuscript, or in the text or in the figure legends. An alternative way can be also to separately display tables with the means ± SEMs (unit of measurement) as Supplementary material.
Response: Thank you for pointing this out. We have added means ± SEMs for each group of data in the Supplementary Table S2.
Supplementary data: if results are not significant, it is not necessary to report the statistical significance applied. Just report the statistical method used. This information should be rather included in the Method section - Data and Statistical Analysis.
Response: Thank you for pointing this out. We have revised the figure legends in the supplementary data. Statistical method has been added in the Data and Statistical Analysis
Methods
First of all: the number of animals employed is elevated. The experimental design does not unambiguously explain the issue. The authors must attest this number or provide more explanation about that, in relation to the statistics applied and the targets of the study.
Response: Thank you for pointing this out. In the experiment 1, we performed behavioral and biochemical assays on 40 rats. In the experiments 2 and 3, we performed behavioral test, WB, and HPLC-MS/MS on 24 PrL administered rats and 24 IL administered rats, respectively. Since ELISA, immunofluorescence and Golgi staining require different treatments of tissues, we performed the above biochemical assays on the other three independent experiments, respectively. In summary we used a total of 232 rats in this study [40 + (24 + 24) × 4 = 232]. We have revised this paragraph as follow:
The experimental design is presented in Figure 8A. The experiment 1 was carried out to monitor behavioral changes after STZ injection through a battery of behavioral tests, and experiment 2 was performed to evaluate molecular and cellular changes in the brain at the time of the onset of depression-like signs. In experiment 1, five rats with surgery-related infections were excluded. Then, 40 rats were randomly grouped into 5 groups (8 animals in each group): (1) Control: no surgery; (2) Vehicle: ICV-aCSF; (3) ICV-STZ 7 d; (4) ICV-STZ 14 d; and (5) ICV-STZ 21 d. Behavioral tests performed as previously de-scribed [76, 77] were conducted on d 7, d 14 and d 21, respectively. The open field test (OFT) was performed at 9:00 AM, and the sucrose preference test (SPT) was performed at 12:00 PM. Following a break the forced swim test (FST) started at 4:00 PM. The novel object recognition test (NORT) was performed at 9:00 AM on d 8, d 15, and d 22, respectively. In experiment 2, 24 rats were randomly grouped into 4 groups (6 animals in each group) for intra-PrL microinjection experiment: (1) Vehicle (ICV)+Vehicle (PrL); (2) Vehicle (ICV)+1-MT (PrL); (3) STZ (ICV)+Vehicle (PrL); (4) STZ (ICV)+1-MT (PrL). In experiment 3, 24 rats were randomly grouped into 4 groups (6 animals in each group) for intra-IL mi-croinjection experiment: (1) Vehicle (ICV)+Vehicle (IL); (2) Vehicle (ICV)+1-MT (IL); (3) STZ (ICV)+Vehicle (IL); (4) STZ (ICV)+1-MT (IL). After the surgical intervention, animal body weight and food consumption were measured daily. The water intake was measured dai-ly from d 1 to d 5 before food and water deprivation. All behavioral tests were conducted on d 7. The OFT was performed at 9:00 AM, and the SPT was performed at 12:00 PM. The novelty-suppressed feeding test (NSFT) was performed at 2:00 PM. Then, the rats were al-lowed to eat and drink freely until the FST started at 4:00 PM. The rats were decapitated on d 8, and the tissue was used for HPLC‒MS/MS and Western blot. The other three inde-pendent experiments identical to experiment 2 and 3 were used for ELISA, IDO activity, immunofluorescence staining and Golgi staining, respectively.
4.5. Experimental design: I suggest to move this section after the section 4.1. Animals. This is important to immediately inform readers, for the sake of clarity.
Response: Thank you for pointing this out. Experimental design section was moved after the section 4.1.
The chromatographic procedure used to separate and measure Trp, 5-HT and Kyn metabolites still needs further clarification. First: has the method described in the paper been validated before? Is there a literature reference? If this is the case, the authors must cite it, eventually indicating modifications employed. Otherwise, the chromatographic conditions presented appear still incomplete, so that the authors should provide more details about that: for instance, the working pressure values; they should also explain the method’s resolution and the theoretical plate specifics in respect to the reported low flow rate and HPLC conditions. The calibration line concentrations and IS retention times are missing. The Supplementary material should include some typical chromatograms.
Response: Thank you for pointing this out. We mainly referred to and cited the detection method of Han et al. and this was reflected in the second line of 4.7.
â‘ Under the condition of the initial mobile phase and flow rate, the working pressure values was around 170 bar (about 2466 psi), and the working pressure values gradually decreased as the proportion of acetonitrile in the mobile phase increased. â‘¡ We usually use the ratio of the difference between the retention time of two adjacent component peaks and the half of the sum of the baseline widths of the two component peaks to evaluate the chromatographic resolution: where tR(1), tR(2) are the retention times of the two components, respectively; y(1), y(2) are the widths of the baseline for corresponding components. When R ≥ 1.5, the two peaks are defined to be completely separated. But in our work we have a triple quadrupole mass spectrometer combining with the chromatographic apparatus, which recognizing and detecting analytes not only rely on retention time, but also through specific ion pairs. For example, although retention time of Trp and IS was close, they could be unambiguous separated by different ion pairs (205.1 for 146.1, and 200 for 154, respectively), therefore, there was no interfere between each other. â‘¢ The theoretical plate of the chromatographic column is approximately 6000. For example, the retention of the IS (negative) is 3.47, the half peak width is 0.1042, and the theoretical plate is 6143. â‘£ Based on the results of the pre-experiment results, we determined the corresponding concentration ranges that could cover the neurotransmitters and metabolites in the samples, and configured standard curves with different concentrations, with the specific parameters listed in the table below:
|
|
Mode |
Conc. STD (ng/ml) |
Parent ion |
Daughter ion |
Retention |
DP (V) |
CE (V) |
|
Trp |
Positive |
10-10000 |
205.1 |
146.1 |
3.53 |
40 |
24 |
|
Kyn |
Positive |
1-1000 |
209.1 |
94.1 |
1.95 |
60 |
22 |
|
KA |
Negative |
1-1000 |
188.0 |
144.0 |
3.68 |
-40 |
-20 |
|
3-HK |
Positive |
10-10000 |
225.1 |
208.1 |
0.89 |
45 |
14 |
|
5-HT |
Positive |
10-10000 |
192.1 |
160.1 |
2.35 |
35 |
12 |
|
IS |
Positive |
100 |
200.0 |
154.0 |
3.49 |
60 |
20 |
|
IS |
Negative |
100 |
198.0 |
181.0 |
3.47 |
-60 |
-20 |
All the above details were added in the revised manuscript and Supplementary data as follow:
4.7. Estimation of serotonin and kynurenine metabolites
The concentrations of serotonin (5-hydroxytryptamin, 5-HT), Trp, Kyn, 3-HK, and KA were measured using an HPLC–MS/MS method as described by Han et al. [83], with slight modifications. Briefly, tissue was transferred into a new EP tube, then mixed with 90 μL of prechilled (4 °C) methanol (0.1% formic acid)-aqueous (8:2, v/v) mixture and 10 μL of IS (2-Cl-Phe; 1 μg/mL). The mixture was homogenized using an ultrasonic homogenizer fol-lowed by centrifugation at 20,000 ×g for 20 min at 4 °C. After centrifugation, the separated supernatant was transferred into a 2 mL auto-sampler, and 10 μL was injected into the system at a flow rate of 0.4 mL/min. The standard curve was prepared using the same procedure as the brain sample. Each data point comes from an individual rat.
The HPLC–MS/MS system consisted of a Dionex UltiMate 3000 HPLC system (Ther-mo, San Jose, CA, USA) and an API 4000Q Trap mass spectrometer (AB SCIEX, Foster City, USA) equipped with an electrospray ionization (ESI) source interface. The optimized mass spectrometric parameters were set as follows: curtain gas, 15 psi; collision gas, 2; ion spray voltage, 5500 V for positive mode or -4500 V for negative mode; ion source tempera-ture, 600 °C; ion source gas 1, 55 psi; ion source gas 2, 55 psi. Accurate quantification was operated in the multiple reaction monitoring (MRM) mode; the transitions were m/z 177.1→160.1 for 5-HT (positive), m/z 205.1→146.1 for Trp (positive), m/z 209.1→94.1 for Kyn (positive), m/z 225.1→208.1 for 3-HK (positive), m/z 188.0→144.0 for KA (negative), m/z 200.0→154.0 for IS (positive), m/z 198.0→181.0 for IS (negative). The declustering potential (DP) was set at 35, 40, 60, 45 and -40 V, and the collision energy (CE) was 12, 24, 22, 14 and -22 V for 5-HT, Trp, Kyn, 3-HK and KA, respectively. Under the condition of the initial mobile phase and flow rate, the working pressure values was around 170 bar (about 2466 psi), and the working pressure values gradually decreased as the proportion of acetonitrile in the mobile phase increased.
Chromatographic separation was performed on an Ultimate XB-C18 column (100 mm×2.1 mm, 5 μm, Welch Materials, Inc.). The theoretical plate of the chromatographic column is approximately 6000. Mobile phase A was water containing 0.1% formic acid, and mobile phase B was acetonitrile. The temperature of the auto-sampler was 4 °C. Gradient separation was set as follows: 0-1 min, 5% B; 1-3 min, 5-60% B; 3.1-5 min, 5% B for column equilibration. The analysis was performed for a total run time of 5 min. Under these conditions, the retention times were 2.35, 3.53, 1.95, 0.89, 3.68, 3.49 and 3.47 min for 5-HT, Trp, Kyn, 3-HK, KA, IS (positive) and IS (negative), respectively. Detailed parameters and typical chromatograms were shown in the Supplementary Table S1 and Supplementary Figure S5. The above data were recorded and analyzed using AB SCIEX Analyst 1.6 software.
5-hydroxytryptamin (5-HT, purity≥98%, Cat# H9523), Tryptophan (Trp, purity≥99.5%, Cat# 93659), kynurenine (Kyn, purity≥98%, Cat# K8625), 3-hydroxy-DL-kynurenine (3-HK, purity≥98%, Cat# 148776) and kynurenic acid (KA, purity≥98%, Cat# K3375) were pur-chased from Sigma‒Aldrich (St. Louis, MO, USA). 2-Chloro-L-phenylalanine (2-Cl-Phe, purity=98%, Cat# C105993), used as an internal standard (IS) was purchased from Alad-din Inc. (Shanghai, China). HPLC-grade acetonitrile and methanol were obtained from Fisher Chemical (Fisher Scientific, Shanghai, China).
These same questions (has the method described in the paper been validated before? Is there a literature reference?) are valuable for all the other techniques and methodologies used in this work, except for the commercially purchased ones. Please verify.
Response: Thank you for pointing this out. All experimental methods in this study have been previously validated and relevant references have been cited in the manuscript.

Reviewer 2 Report (New Reviewer)
Comments and Suggestions for Authors
I want to thank the authors for their reply. The manuscript improved.
Comments on the Quality of English Languagestill there are some spelling mistakes, so please revise
Author Response
Dear Reviewer:
We checked and revised all spelling mistakes in the manuscript which we hope meet with approval.
Reviewer 3 Report (Previous Reviewer 1)
Comments and Suggestions for Authors
In the present form the manuscript is suitable for publication.
Author Response
Dear Reviewer:
Thanks again for your valuable suggestions on our manuscript.
Round 3
Reviewer 1 Report (New Reviewer)
Comments and Suggestions for Authors
Thank you for your prompt replies and for uploading this version of the manuscript, accordingly to my additional comments. This version of the manuscript ijms-3010257 is now clearly presented in most of its sections. An Abbreviation paragraph has been also added, as well as Supplementary materials have been improved, as suggested.
I have few suggestions for authors and, for that, I just ask for still some effort:
1- Abstracts
page 1, line 25: "neurotoxic", change into "neurotoxicogenic" or “potentially neurotoxic” or similar expressions- indeed, when introducing these pathways, it should be considered that they are all physiological branches, and only abnormal substrate production/accumulation or dysfunctional metabolic fluxes can lead to neurotoxicity through some specific substrates, as quinolinic acid. This revision point is not mandatory, but in my opinion is more correct.
Discussion:
Page 14-15, lines 373-379, modify as follows: These findings require deeper investigation, they would reflect region-dependent Trp metabolism balances, also suggesting that AD-related depression may have a unique pathogenesis.
Methods
This section has been improved. Anyway, these points still require to be addressed.
- The number of rats employed is high. The authors should more clearly indicate in the text how they attained this number. If they replicated the experiments four times (is this the case?), this should be present in the text of the manuscript and explained, for instance on the basis of statistical design or methods' variability.
- The chromatographic method used in the study has been modified from that previously reported by Han and coauthors [84]. Anyway, there are still some points scarcely presented and reported about the experimental conditions adopted here. Running times are quite short for a 5 um particle column, at low mobile phase fluxes. Is this information correct? Please verify. Since chromatographic conditions in the present and the past study do not fully match and the investigated analytes are almost different, I still suggest that the supplementary material should contain a typical chromatogram, also providing the analytes' concentrations obtained. For the same reasons: some information about the method's features of linearity, precision, accuracy and sensitivity must be present in the method section. This is important, since you measured the Kyn/Trp ratios as indices of IDO activity, which is the key point of the study.
These are overall minor points, but the last two methodological ones are instead important, and they should be better addressed and focused.
Comments on the Quality of English Language
I suggest that the Authors always revise their manuscript latest version.
Author Response
Please see the attachment.

This manuscript is a resubmission of an earlier submission. The following is a list of the peer review reports and author responses from that submission.
Round 1
Reviewer 1 Report
Comments and Suggestions for Authors
The submitted manuscript aimed to investigate the potential role of IDO and some kynurenines in depression-like behavior induced in rats by administration of streptozocin. Although interesting, the manuscript requires a number of changes prior to publication:
1) Introduction
To figure out the main locations, we detected IDO-related metabolisms in several brain regions when these ICV-STZ rats exhibited depression-like behaviors. It was impressive (???) to show that prelimbic cortex (PrL) and infralimbic cortex (IL) were the most obvious regions where IDO was significantly activated by STZ exposure. - correct language and, if this is authors' observation, please provide ref. nr.
2) Discussion - the authors observed that 1-MT significantly increased glial marker GFAP expression in the IL - what is the explanation? At some points the authors state that the antidepressant effects of 1-MT in the IL might be at least partially mediated by local anti-inflammatory processes and the blockade of the neurotoxicity branches. It contrasts with observation on an increased GFAP.
Comments on the Quality of English Language
Language requires intense editing. SOME of the examples:
"Evidence suggests that depression might be a prodromal symptom of AD and
seriously reduce quality of life [3, 4], while the pathophysiological mechanisms that underlie depression in AD patients are less well defined. These patients were treated with the conventional antidepressants, but the efficacy is uncertain and even serious adverse effects have been observed [5, 6]. These facts suggest that the pathological mechanisms of depression in AD patients may be different from that of major depressive disorder, and it is necessary to be elucidated. " - complete rewriting necessary
to change "intermediates of Kyn are metabolized into kynurenic acid (KA), which has neuroprotective effects mainly produced by kynurenine aminotransferase (KAT) II located in astrocytes, while Kyn is also metabolized in microglia by kynurenine 3-monooxygenase (KMO) to 3-hydroxykynurenine (3-HK) and quinolinic acid to induce excitotoxic effects"
to change - It has been reported that the abnormalities in IDO-related Kyn metabolism occurred in glial play a prominent role...
to change - Basically, central KAT II is predominantly located in astrocytes and metabolites Kyn to KA, whereas KMO is majorly expressed in microglia and metabolites Kyn to 3-HK
Please, change description in methods, lines 660-665; at present it is an instruction and not the description of what was done
Author Response
Dear Reviewer:
Thank you for your comments concerning our manuscript entitled “Inhibition of indoleamine 2,3-dioxygenase in the prelimbic and infralimbic cortices exerts antidepressant effects through different pathways in ICV-STZ rats” (ID: ijms-2902463). Those comments are all valuable and very helpful for revising and improving our paper, as well as the important guiding significance to us research. We have studied comments carefully and have made correction which we hope meet with approval. Revised portion are marked in red in the manuscript. The main corrections in the paper and the responds to the reviewer's comments are as follow.
Comments 1: Introduction-To figure out the main locations, we detected IDO-related metabolisms in several brain regions when these ICV-STZ rats exhibited depression-like behaviors. It was impressive (???) to show that prelimbic cortex (PrL) and infralimbic cortex (IL) were the most obvious regions where IDO was significantly activated by STZ exposure. - correct language and, if this is authors' observation, please provide ref. nr.
Response 1: Thank you for pointing this out. This part of the description is based on our results in section 2.2. We revised these sentences as follow:
IDO-related metabolisms in several brain regions were detected when ICV-STZ rats exhibited depression-like behaviors (Figure 2). And the results showed that prelimbic cortex (PrL) and infralimbic cortex (IL) were the most obvious brain regions where IDO was significantly activated by STZ exposure.
Comments 2: Discussion - the authors observed that 1-MT significantly increased glial marker GFAP expression in the IL - what is the explanation? At some points the authors state that the antidepressant effects of 1-MT in the IL might be at least partially mediated by local anti-inflammatory processes and the blockade of the neurotoxicity branches. It contrasts with observation on an increased GFAP.
Response 2: As you are concerned, we did observe this phenomenon, although it is also not clear why intra-IL injection of 1-MT significantly increased the expression of GFAP. In addition, although the expression of GFAP was significantly increased in the IL, neither the number and morphology of astrocytes nor the expression of glial cell transporter proteins were changed. So, we thought that the antidepressant effect of intra-IL injection of 1-MT may still be mainly through local anti-inflammatory processes and the blockade of the neurotoxicity branches. We also added description at the end of the third paragraph of the discussion section as follow: Although we did not observe a decrease in GFAP levels in the IL of ICV-STZ rats, surprisingly, intra-IL injection of 1-MT significantly increased the expression of GFAP in both the control and model groups, while other astrocyte-related measures did not change significantly. The changes of GFAP induced by 1-MT in IL and its biological mechanism are still unclear and need further study.
Response to Comments on the Quality of English Language
Point 1: "Evidence suggests that depression might be a prodromal symptom of AD and seriously reduce quality of life [3, 4], while the pathophysiological mechanisms that underlie depression in AD patients are less well defined. These patients were treated with the conventional antidepressants, but the efficacy is uncertain and even serious adverse effects have been observed [5, 6]. These facts suggest that the pathological mechanisms of depression in AD patients may be different from that of major depressive disorder, and it is necessary to be elucidated. "
Response 1: Evidence suggests that depression may serve as a prodromal symptom of AD and significantly diminish the quality of life, while the pathophysiological mechanisms underlying depression in AD patients remain less well defined.
Point 2: "intermediates of Kyn are metabolized into kynurenic acid (KA), which has neuroprotective effects mainly produced by kynurenine aminotransferase (KAT) II located in astrocytes, while Kyn is also metabolized in microglia by kynurenine 3-monooxygenase (KMO) to 3-hydroxykynurenine (3-HK) and quinolinic acid to induce excitotoxic effects"
Response 2: Subsequently, Kyn is metabolized by kynurenine aminotransferase (KAT) II in astrocytes, and converted to kynurenic acid (KA), which exerts neuroprotective effects. In addition, Kyn is metabolized by kynurenine 3-monooxygenase (KMO) in microglia, and its metabolites the 3-hydroxykynurenine (3-HK) and quinolinic acid have a potential to induce excitotoxic effects.
Point 3: It has been reported that the abnormalities in IDO-related Kyn metabolism occurred in glial play a prominent role...
Response 3: The abnormalities in IDO-related Kyn metabolism occurring in glial cells have been reported to play a prominent role in the processes of neuroinflammation.
Point 4: Basically, central KAT II is predominantly located in astrocytes and metabolites Kyn to KA, whereas KMO is majorly expressed in microglia and metabolites Kyn to 3-HK
Response 4: Essentially, central KAT II is predominantly localized within astrocytes and contributes to the metabolism of Kyn to KA, while KMO is expressed mainly in microglia and facilitates the conversion of Kyn to 3-HK.
Point 5: Please, change description in methods, lines 660-665; at present it is an instruction and not the description of what was done
Response 5: Subsequently, the supernatant was collected and the reagents or samples were added to a 96-well plate as directed and incubated at 37 °C for 45 minutes in the dark. Then, 50 μL of Fluorescent Developer Solution were added to each well and incubated for 3 hours at 45 °C in the dark with gentle shaking. Finally, the plates were cooled to room temperature for 1 h and fluorescence was measured in end-point mode (λex/nm = 402 /λem/nm = 488).
Reviewer 2 Report
Comments and Suggestions for Authors
Dear authors;
AD comorbid depressive disorders are a majority problem for patients; thus, it is important to study and find new models of these diseases. Your manuscript is written up to the above- mentioned problem. The manuscript is readable and well-organized for potential readers. All experiments are indicated properly, nevertheless I have few questions/suggestions:
1. You did Western blot just in d7 to show that AD wasn’t developed in those periods. Why don’t you confirmed AD on d14 and d21, to show that not just behaviorally but also molecularly AD was developing in ICV-STZ model
2. For the behavioral part—in your experimental design almost all behavior tests were induced in one day (OFT, SPT, FST), isn’t too overwhelming for them? Is in NSF at 2:00 PM with SPT two hours before (12:00 PM) time for sucrose consumption that does not interfere with latency?? I know that for you time for decapitation was crucial (d8), but you can really trust tests conducted at the end of a “busy” day.
Thank you for your response.
Author Response
Dear Reviewer:
Thank you for your comments concerning our manuscript entitled “Inhibition of indoleamine 2,3-dioxygenase in the prelimbic and infralimbic cortices exerts antidepressant effects through different pathways in ICV-STZ rats” (ID: ijms-2902463). Those comments are all valuable and very helpful for revising and improving our paper, as well as the important guiding significance to us researches. We have studied comments carefully and have made correction which we hope meet with approval. Revised portion are marked in red in the manuscript. The main corrections in the paper and the responds to the reviewer's comments are as follow.
Comments 1: You did Western blot just in d7 to show that AD wasn’t developed in those periods. Why don’t you confirmed AD on d14 and d21, to show that not just behaviorally but also molecularly AD was developing in ICV-STZ model.
Response 1: Thank you for pointing that out. We have found in previous studies (PMID: 29080930 and 33352241) that ICV-STZ rats began to show cognitive impairment in learning and memory at d14, and showed the characteristic pathological changes of AD in the hippocampus with increased Aβ expression and elevated Tau phosphorylation. Therefore, in this paper, we did not review the AD-related molecular biological alterations on d14 and d21.
Comments 2: For the behavioral part—in your experimental design almost all behavior tests were induced in one day (OFT, SPT, FST), isn’t too overwhelming for them? Is in NSF at 2:00 PM with SPT two hours before (12:00 PM) time for sucrose consumption that does not interfere with latency?? I know that for you time for decapitation was crucial (d8), but you can really trust tests conducted at the end of a “busy” day.
Response 2: We feel great thanks for your professional review work on this problem. Before the SPT and NSFT, the rats were deprived of food and water for 24 h. As you are concerned, although drinking sucrose may alleviate the hunger of the rats to a certain extent on the test day, we set up a vehicle group of rats in each behavioral experiment, so we can exclude the interference and bias to a certain extent. In addition, we only conduct behavioral experiments on 2 rats of each group on the same day, and each of our behavioral equipment can guarantee 4 animals to be tested at the same time. All behavioral experiments were completed on the same day.
Round 2
Reviewer 1 Report
Comments and Suggestions for Authors
The manuscript is now suitable for publication
Author Response
Dear Reviewer:
We thank the reviewer for the professional and constructive comments regarding our manuscript.